# Does progress on ImageNet transfer to real-world datasets?

**Alex Fang**
University of Washington
apf1@cs.washington.edu

**Simon Kornblith**[*]
Google Research, Brain Team
skornblith@google.com

**Ludwig Schmidt**[*]
University of Washington, AI2
schmidt@cs.washington.edu

## Abstract

Does progress on ImageNet transfer to real-world datasets? We investigate this question by evaluating ImageNet pre-trained models with varying accuracy (57% - 83%) on six practical image classification datasets. In particular, we study datasets collected with the goal of solving real-world tasks (e.g., classifying images from camera traps or satellites), as opposed to web-scraped benchmarks collected for comparing models. On multiple datasets, models with higher ImageNet accuracy do not consistently yield performance improvements. For certain tasks, interventions such as data augmentation improve performance even when architectures do not. We hope that future benchmarks will include more diverse datasets to encourage a more comprehensive approach to improving learning algorithms.

## 1 Introduction

ImageNet is one of the most widely used datasets in machine learning. Initially, the ImageNet competition played a key role in re-popularizing neural networks with the success of AlexNet in 2012. Ten years later, the ImageNet dataset is still one of the main benchmarks for state-of-the-art computer vision models [35, 61, 20, 37, 24, 69, 52]. As a result of ImageNet's prominence, the machine learning community has invested tremendous effort into developing model architectures, training algorithms, and other methodological innovations with the goal of increasing performance on ImageNet. Comparing methods on a common task has important benefits because it ensures controlled experimental conditions and results in rigorous evaluations. But the singular focus on ImageNet also raises the question whether the community is over-optimizing for this specific dataset.

As a first approximation, ImageNet has clearly encouraged effective methodological innovation beyond ImageNet itself. For instance, the key finding from the early years of ImageNet was that large convolution neural networks (CNNs) can succeed on contemporary computer vision datasets by leveraging GPUs for training. This paradigm has led to large improvements in other computer vision tasks, and CNNs are now omnipresent in the field. Nevertheless, this clear example of transfer to other tasks early in the ImageNet evolution does not necessarily justify the continued focus ImageNet still receives. For instance, it is possible that early methodological innovations transferred more broadly to other tasks, but later innovations have become less generalizable. The goal of our paper is to investigate this possibility specifically for neural network architecture and their transfer to real-world data not commonly found on the Internet.

When discussing the transfer of techniques developed for ImageNet to other datasets, a key question is what other datasets to consider. Currently there is no comprehensive characterization of the many machine learning datasets and transfer between them. Hence we restrict our attention to a limited but well-motivated family of datasets. In particular, we consider classification tasks derived from image data that were specifically collected with the goal of classification in mind. This is in contrast to many standard computer vision datasets – including ImageNet – where the constituent

---

[*]Equal contribution

37th Conference on Neural Information Processing Systems (NeurIPS 2023) Track on Datasets and Benchmarks.

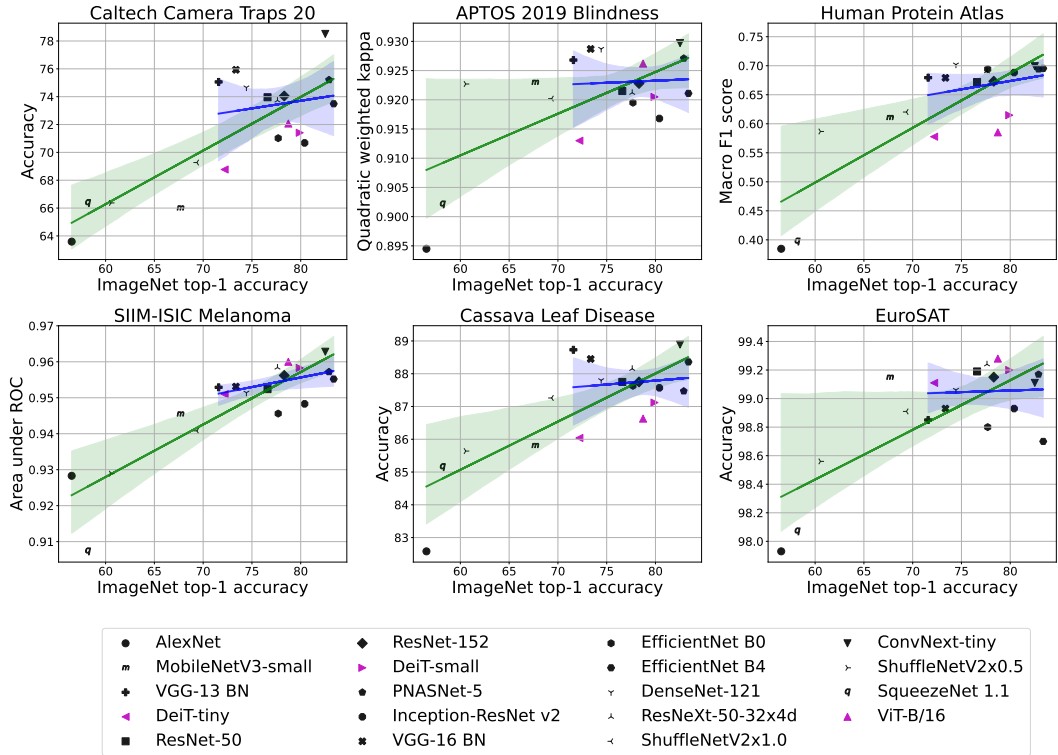

Figure 1: Overview of transfer performance across models from ImageNet to each of the datasets we study. Although there seems to be a strong linear trends between ImageNet accuracy and the target metrics (green), these trends become less certain when we restrict the models to those above 70% ImageNet accuracy (blue). Versions with error bars and spline interpolation can be found in Appendix B.

images were originally collected for a different purpose, posted to the web, and later re-purposed for benchmarking computer vision methods. Concretely, we study six datasets ranging from leaf disease classification over melanoma detection to categorizing animals in camera trap images. Since these datasets represent real-world applications, transfer of methods from ImageNet is particularly relevant.

We find that on four out of our six real-world datasets, ImageNet-motivated architecture improvements after VGG resulted in little to no progress (see Figure 1). Specifically, when we fit a line to downstream model accuracies as a function of ImageNet accuracy, the resulting slope is less than 0.05. The two exceptions where post-VGG architectures yield larger gains are the Caltech Camera Traps-20 (CCT-20) [3] dataset (slope 0.11) and the Human Protein Atlas Image Classification [47] dataset (slope 0.29). On multiple other datasets, we find that task-specific improvements such as data augmentations or extra training data lead to larger gains than using a more recent ImageNet architecture. We evaluate on a representative testbed of 19 ImageNet models, ranging from the seminal AlexNet [35] over VGG [61] and ResNets [20] to the more recent and higher-performing EfficientNets [66] and ConvNexts [39] (ImageNet top-1 accuracies 56.5% to 83.4%). Our testbed includes three Vision Transformer models to cover non-CNN architectures.

Interestingly, our findings stand in contrast to earlier work that investigated image classification benchmarks such as CIFAR-10 [34], PASCAL VOC 2007 [14], and Caltech-101 [15] that were scraped from the Internet. On these datasets, Kornblith et al. [31] found consistent gains in downstream task accuracy for a similar range of architectures as we study in our work. Taken together, these findings indicate that ImageNet accuracy is a good predictor for other web-scraped datasets, but less informative for many real-world image classification datasets that are not sourced through the web. On the other hand, the CCT-20 data point shows that even very recent ImageNet models do help on some downstream tasks that do not rely on images from the web. Overall, our results highlight the need for a more comprehensive understanding of machine learning datasets to build and evaluate broadly useful data representations. We provide sample training code and dataset information at `https://github.com/mlfoundations/imagenet-applications-transfer` to ensure reproducibility and encourage future research in this direction.

## 2 Related Work

**Transferability of ImageNet architectures.** Although there is extensive previous work investigating the effect of architecture upon the transferability of ImageNet-pretrained models to different datasets, most of this work focuses on performance on datasets collected for the purpose of benchmarking. Kornblith et al. [31] previously showed that ImageNet accuracy of different models is strongly correlated with downstream accuracy on a wide variety of web-scraped object-centric computer vision benchmark tasks. Later studies have investigated the relationship between ImageNet and transfer accuracy for self-supervised networks [13, 33, 46], adversarially trained networks [56], or networks trained with different loss functions [32], but still evaluate primarily on web-scraped benchmark tasks. The Visual Task Adaptation Benchmark (VTAB) [81] comprises a more diverse set of tasks, including natural and non-natural classification tasks as well as non-classification tasks, but nearly all consist of web-scraped or synthetic images. In the medical imaging domain, models have been extensively evaluated on real-world data, with limited gains from newer models that perform better on ImageNet [53, 5, 29].

Most closely related to our work, Tuggener et al. [71] investigate performance of 500 CNN architectures on yet another set of datasets, several of which are not web-scraped, and find that accuracy correlates poorly with ImageNet accuracy when training from scratch, but correlations are higher when fine-tuning ImageNet-pretrained models. Our work differs from theirs in our focus solely on real-world datasets (e.g., from Kaggle competitions) and in that we perform extensive tuning in order to approach the best single-model performance obtainable on these datasets whereas Tuggener et al. [71] instead devote their compute budget to increasing the breadth of architectures investigated.

**Transferability of networks trained on other datasets.** Other work has evaluated transferability of representations of networks trained on datasets beyond ImageNet. Most notably, Abnar et al. [1] explore the relationship between upstream and downstream accuracy for models pretrained on JFT and ImageNet-21K and find that, on many tasks, downstream accuracy saturates with upstream accuracy. However, they evaluate representational quality using linear transfer rather than end-to-end fine-tuning. Other studies have investigated the impact of relationships between pretraining and fine-tuning tasks [80, 43] or the impact of scaling the model and dataset [16, 30].

Another direction of related work relates to the effect of pretraining data on transfer learning. Huh et al. [26] look into the factors that make ImageNet good for transfer learning. They find that fine-grained classes are not needed for good transfer performance, and that reducing the dataset size and number of classes only results in slight drops in transfer learning performance. Though there is a common goal of exploring what makes transfer learning work well, our work differs from this line of work by focusing on the fine-tuning aspect of transfer learning.

**Other studies of external validity of benchmarks.** Our study fits into a broader literature investigating the external validity of image classification benchmarks. Early work in this area identified lack of diversity as a key shortcoming of the benchmarks of the time [51, 68], a problem that was largely resolved with the introduction of the much more diverse ImageNet benchmark [9, 55]. More recent studies have investigated the extent to which ImageNet classification accuracy correlates with accuracy on out-of-distribution (OOD) data [54, 67] or accuracy as measured using higher-quality human labels [57, 70, 4].

As in previous studies of OOD generalization, transfer learning involves generalization to test sets that differ in distribution from the (pre-)training data. However, there are also key differences between transfer learning and OOD generalization. First, in transfer learning, additional training data from the target task is used to adapt the model, while OOD evaluations usually apply trained models to a new distribution without any adaptation. Second, OOD evaluations usually focus on settings with a shared class space so that evaluations without adaptation are possible. In contrast, transfer learning evaluation generally involves downstream tasks with classes different from those in the pretraining dataset. These differences between transfer learning and OOD generalization are not only conceptual but also lead to different empirical phenomena. Miller et al. [44] has shown that in-distribution accuracy improvements often directly yield out-of-distribution accuracy improvements as well. This is the opposite of our main experimental finding that ImageNet improvements do not directly yield performance improvements on many real-world downstream tasks. Hence our work demonstrates an important difference between OOD generalization and transfer learning.

# 3 Datasets

As mentioned in the introduction, a key choice in any transfer study is the set of target tasks on which to evaluate model performance. Before we introduce our suite of target tasks, we first describe three criteria that guided our dataset selection: (i) diverse data sources, (ii) relevance to an application, and (iii) availability of well-tuned baseline models for comparison.

## 3.1 Selection criteria

Prior work has already investigated transfer of ImageNet architectures to many downstream datasets [10, 58, 6, 61]. The 12 datasets used by Kornblith et al. [31] often serve as a standard evaluation suite (e.g., in [56, 13, 52]). While these datasets are an informative starting point, they are all object-centric natural image datasets, and do not represent the entire range of image classification problems. There are many applications of computer vision; the Kaggle website alone lists more than 1,500 datasets as of May 2022. To understand transfer from ImageNet more broadly, we selected six datasets guided by the following criteria.

**Diverse data sources.** Since collecting data is an expensive process, machine learning researchers often rely on web scraping to gather data when assembling a new benchmark. This practice has led to several image classification datasets with different label spaces such as food dishes, bird species, car models, or other everyday objects. However, the data sources underlying these seemingly different tasks are actually often similar. Specifically, we surveyed the 12 datasets from Kornblith et al. [31] and found that all of these datasets were harvested from the web, often via keyword searches in Flickr, Google image search, or other search engines (see Appendix J). This narrow range of data sources limits the external validity of existing transfer learning experiments. To get a broader understanding of transfer from ImageNet, we focus on scientific, commercial, and medical image classification datasets that were not originally scraped from the web.

**Application relevance.** In addition to the data source, the classification task posed on a given set of images also affects how relevant the resulting problem is for real-world applications. For instance, it would be possible to start with real-world satellite imagery that shows multiple building types per image, but only label one of the building types for the purpose of benchmarking (e.g., to avoid high annotation costs). The resulting task may then be of limited value for an actual application involving satellite images that requires all buildings to be annotated. We aim to avoid such pitfalls by limiting our attention to tasks that were assembled by domain experts with a specific application in mind.

**Availability of baselines.** If methodological progress does not transfer from ImageNet to a given target task, we should expect that, as models perform better on ImageNet, accuracy on the target task saturates. However, observing such a trend in an experiment is not sufficient to reach a conclusion regarding transfer because there is an alternative explanation for this empirical phenomenon. Besides a lack of transfer, the target task could also simply be easier than the source task so that models with sub-optimal source task accuracy already approach the Bayes error rate. As an illustrative example, consider MNIST as a target task for ImageNet transfer. A model with mediocre ImageNet accuracy is already sufficient to get 99% accuracy on MNIST, but this finding does not mean that better ImageNet models are insufficient to improve MNIST accuracy — the models have already hit the MNIST performance ceiling.

More interesting failures of transfer occur when ImageNet architectures plateau on the target task, but it is still possible to improve accuracy beyond what the best ImageNet architecture can achieve without target task-specific modifications. In order to make such comparisons, well-tuned baselines for the target task are essential. If improving ImageNet accuracy alone is insufficient to reach these well-tuned baselines, we can indeed conclude that architecture transfer to this target task is limited. In our experiments, we use multiple datasets from Kaggle competitions since the resulting leaderboards offer well-tuned baselines arising from a competitive process.

## 3.2 Datasets studied

The datasets studied in this work are practical and cover a variety of applications. We choose four of the most popular image classification competitions on Kaggle, as measured by number of competitors, teams, and submissions. Each of these competitions is funded by an organization with the goal of advancing performance on that real-world task. Additionally, we supplement these datasets with Caltech Camera Traps [3] and EuroSAT [21] to broaden the types of applications studied. The datasets we study are all under 50,000 training images, potentially due to the cost of collecting and

Table 1: We examine a variety of real-world datasets that cover different types of tasks.

| Dataset | # of classes | Train size | Eval size | Eval metric | Kaggle |
|---|---|---|---|---|---|
| Caltech Camera Traps | 15 | 14,071 | 15,215 | Accuracy | |
| APTOS 2019 Blindness | 5 | 2,930 | 732 | Quadratic weighted kappa | ✓ |
| Human Protein Atlas | 28 | 22,582 | 5,664 | Macro F1 score | ✓ |
| SIIM-ISIC Melanoma | 2 | 46,372 | 11,592 | Area under ROC | ✓ |
| Cassava Leaf Disease | 5 | 17,118 | 4,279 | Accuracy | ✓ |
| EuroSAT | 10 | 21,600 | 5,400 | Accuracy | |

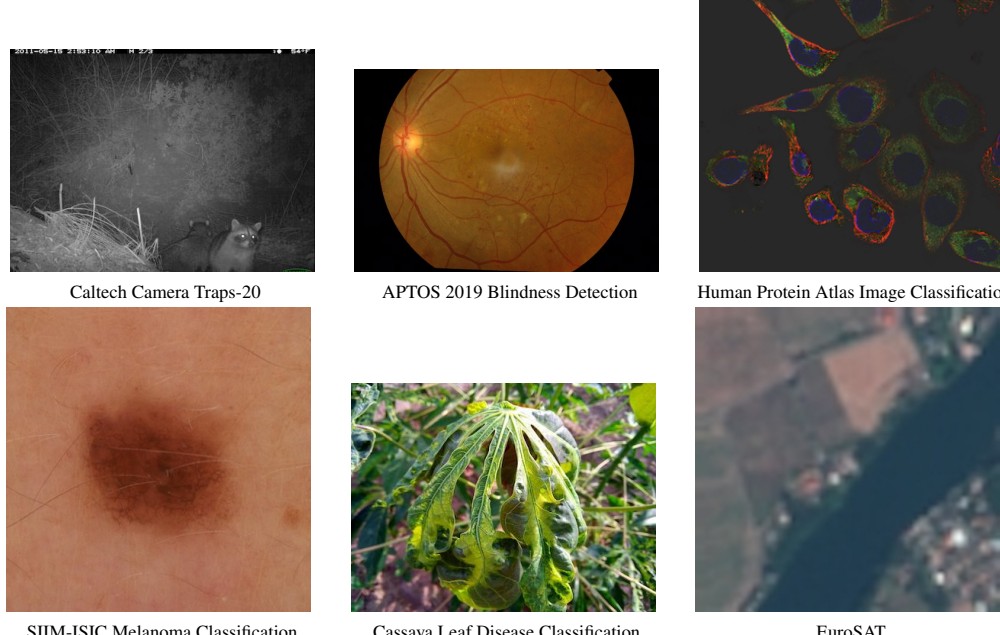

Caltech Camera Traps-20     APTOS 2019 Blindness Detection     Human Protein Atlas Image Classification

SIIM-ISIC Melanoma Classification     Cassava Leaf Disease Classification     EuroSAT

Figure 2: Sample images from each of the datasets.

annotating real-world data, and our focus on transfer from ImageNet limits this study to RGB datasets. Details for each dataset can be found in Table 1 [2].

# 4 Main Experiments

We run our experiments across 19 model architectures, including both CNNs and Vision Transformers (ViT and DeiT). They range from 57% to 83% ImageNet top-1 accuracy, allowing us to observe the relationship between ImageNet performance and target dataset performance.

In order to get the best performance out of each architecture, we do extensive hyperparameter tuning over learning rate, weight decay, optimizer, and learning schedule. Experiment setup details can be found in Appendix C. We now present our results for each of the datasets we investigated. Figure 1 summarizes our results across all datasets, with additional statistics in Table 15. Appendix A contains complete results for all datasets across the hyperparameter grids.

Table 2: We summarize the blue regression lines from Figure 1, calculated on models above 70% ImageNet accuracy, with their correlation and slope. Slope is calculated so that all metrics have a range from 0 to 100.

| Dataset | Correlation | Slope |
|---|---|---|
| Caltech Camera Traps | 0.17 | 0.11 |
| APTOS 2019 Blindness | 0.06 | 0.01 |
| Human Protein Atlas | 0.26 | 0.29 |
| SIIM-ISIC Melanoma | 0.44 | 0.05 |
| Cassava Leaf Disease | 0.12 | 0.02 |
| EuroSAT | 0.05 | 0.00 |

---

[2]Dataset download links and PyTorch datasets and splits can be found at `https://github.com/mlfoundations/imagenet-applications-transfer`.

## 4.1 Caltech Camera Traps

Beery et al. [3] created Caltech Camera Traps-20 (CCT-20) using images taken from camera traps deployed to monitor animal populations. The images contain 15 animal classes, as well as an empty class that we remove for our experiments [3]. The dataset has two sets of validation and test sets which differ by whether they come from locations that are the same as or different from the training set locations. While one of the goals of the dataset is to study generalization to new environments, here we only study the sets from the same locations. Though CCT-20 is not a Kaggle competition, it is a subset of iWildCam Challenge 2018, whose yearly editions have been hosted on Kaggle.

We see in Figure 1 (top-left) an overall positive trend between ImageNet performance and CCT-20 performance. The overall trend is unsurprising, given the number of animal classes present in ImageNet. But despite the drastic reduction in the number of classes when compared to ImageNet, CCT-20 has its own set of challenges. Animals are often pictured at difficult angles, and sometimes are not even visible in the image because a sequence of frames triggered by activity all have the same label. Despite these challenges, an even higher performing model still does better on this task - we train a CLIP ViT L/14-336px model (85.4% ImageNet top-1) with additional augmentation to achieve 83.4% accuracy on CCT-20.

## 4.2 APTOS 2019 Blindness Detection

This dataset was created for a Kaggle competition run by the Asia Pacific Tele-Ophthalmology Society (APTOS) with the goal of advancing medical screening for diabetic retinopathy in rural areas [2]. Images are taken using fundus photography and vary in terms of clinic source, camera used, and time taken. Images are labeled by clinicians on a scale of 0 to 4 for the severity of diabetic retinopathy. Given the scaled nature of the labels, the competition uses quadratic weighted kappa (QWK) as the evaluation metric. We create a local 80% to 20% random class-balanced train/validation split, as the competition test labels are hidden.

We find that models after VGG do not show significant improvement. Similar to in CCT-20, DeiT and EfficientNets performs slightly worse, while deeper models from the same architecture slightly help performance. We also find that accuracy has a similar trend as QWK, despite it being an inferior metric in the context of this dataset.

When performance stagnates, one might ask whether we have reached a performance limit for our class of models on the dataset. To answer this question, we compare with the Kaggle leaderboard's top submissions. The top Kaggle submission achieves 0.936 QWK on the private leaderboard (85% of the test set) [75]. They do this by using additional augmentation, using external data, training on L1-loss, replacing the final pooling layer with generalized mean pooling, and ensembling a variety of models trained with different input sizes. The external data consists of 88,702 images from the 2015 Diabetic Retinopathy Detection Kaggle competition.

Even though performance saturates with architecture, we find that additional data augmentation and other interventions still improve accuracy. We submitted our ResNet-50 and ResNet-152 models with additional interventions, along with an Inception-ResNet v2 [65] model with hyperparameter tuning. We find that increasing color and affine augmentation by itself can account for a 0.03 QWK point improvement. Once we train on 512 input size, additional augmentation, and additional data, our ResNet-50 and Inception-ResNet v2 both achieve 0.896 QWK on the private leaderboard, while ResNet-152 achieves 0.890 QWK, once again suggesting that better ImageNet architectures by themselves do not lead to increased performance on this task.

As a comparison, the ensemble from the top leaderboard entry included a single model Inception-ResNet v2 trained with additional interventions that achieves 0.927 QWK. We submitted the original models we trained to Kaggle as well, finding that the new models trained with additional interventions do at least 0.03 QWK points better. See Appendix F for additional experimental details. Both this result and the gap between our models and the top leaderboard models show that there exist interventions that do improve task performance.

## 4.3 Human Protein Atlas Image Classification

The Human Protein Atlas runs the Human Protein Atlas Image Classification competition on Kaggle to build an automated tool for identifying and locating proteins from high-throughput microscopy

---

[3]Empty class is removed for the classification experiments in Table 1 of Beery et al. [3]

images [47]. Images can contain multiple of the 28 different proteins, so the competition uses the macro F1 score. Given the multi-label nature of the problem, this requires thresholding for prediction. We use a 73% / 18% / 9% train / validation / test-validation split created by a previous competitor [49]. We report results on the validation split, as we find that the thresholds selected for the larger validation split generalize well to the smaller test-validation split.

We find a slightly positive trend between task performance and ImageNet performance, even when ignoring AlexNet and MobileNet. This is surprising because ImageNet is quite visually distinct from human protein slides. These results suggest that models with more parameters help with downstream performance, especially for tasks that have a lot of room for improvement.

Specific challenges for this dataset are extreme class imbalance, multi-label thresholding, and generalization from the training data to the test set. Competitors were able to improve performance beyond the baselines we found by using external data as well as techniques such as data cleaning, additional training augmentation, test time augmentation, ensembling, and oversampling [8, 49, 59]. Additionally, some competitors modified commonly-used architectures by substituting pooling layers or incorporating attention [49, 83]. Uniquely, the first place solution used metric learning on top of a single DenseNet121 [8]. These techniques may be useful when applied to other datasets, but are rarely used in a typical workflow.

## 4.4 SIIM-ISIC Melanoma Classification

The Society for Imaging Informatics in Medicine (SIIM) and the International Skin Imaging Collaboration (ISIC) jointly ran this Kaggle competition for identifying Melanoma [60], a serious type of skin cancer. Competitors use images of skin lesions to predict the probability that each observed image is malignant. Images come from the ISIC Archive, which is publicly available and contains images from a variety of countries. The competition provided 33,126 training images, plus an additional 25,331 images from previous competitions. We split the combined data into an 80% to 20% class-balanced and year-balanced train/validation split. Given the imbalanced nature of the data (8.8% positive), the competition uses area under ROC curve as the evaluation metric.

We find only a weak positive correlation (0.44) between ImageNet performance and task performance, with a regression line with a normalized slope of close to zero (0.05). But if we instead look at classification accuracy, Appendix G shows that there is a stronger trend for transfer than that of area under ROC curve, as model task accuracy more closely follows the same order as ImageNet performance. This difference shows that characterizing the relationship between better ImageNet models and better transfer performance is reliant on the evaluation metric as well. We use a relatively simple setup to measure the impact of ImageNet models on task performance, but we know we can achieve better results with additional strategies. The top two Kaggle solutions used models with different input size, ensembling, cross-validation and a significant variety of training augmentation to create a stable model that generalized to the hidden test set [17, 48].

## 4.5 Cassava Leaf Disease Classification

The Makerere Artificial Intelligence Lab is an academic research group focused on applications that benefit the developing world. Their goal in creating the Cassava Leaf Disease Classification Kaggle competition [42] was to give farmers access to methods for diagnosing plant diseases, which could allow farmers to prevent these diseases from spreading, increasing crop yield. Images were taken with an inexpensive camera and labeled by agricultural experts. Each image was classified as healthy or as one of four different diseases. We report results using a 80%/20% random class-balanced train/validation split of the provided training data.

Once we ignore models below 70% ImageNet accuracy, the relationship between the performance on the two datasets has both a weak positive correlation (0.12) and a near-zero normalized slope (0.02). While these are natural images similar to portions of ImageNet, it is notable that ImageNet contains very few plant classes (e.g., buckeye, hip, rapeseed). Yet based on a dataset's perceived similarity to ImageNet, it is surprising that leaf disease classification is not positively correlated with ImageNet, while the microscopy image based Human Protein Atlas competition is. Our results are supported by Kaggle competitors: the first place solution found that on the private leaderboard, EfficientNet B4 [66], MobileNet, and ViT [12] achieve 89.5%, 89.4%, and 88.8% respectively [19]. Their ensemble achieves 91.3% on the private leaderboard.

Table 3: We examine the effect of pre-training augmentation and fine-tuning augmentation on downstream transfer performance. The model specifies the architecture and pre-training augmentation, while each column specifies the downstream task and fine-tuning augmentation. We find that augmentation strategies that improve ImageNet accuracy do not always improve accuracy on downstream tasks. Pre-trained augmentation models are from Wightman et al. [72].

| Model | ImageNet Acc | CCT-20 Base Aug | CCT-20 AugMix | CCT-20 RandAug | APTOS Base Aug | APTOS AugMix | APTOS RandAug |
|---|---|---|---|---|---|---|---|
| ResNet-50 | 76.1 | 72.02 | 72.24 | 73.57 | 0.9210 | 0.9212 | 0.9250 |
| ResNet-50 w/ AugMix | 77.5 | 71.63 | 71.53 | 72.39 | 0.9239 | 0.9152 | 0.9222 |
| ResNet-50 w/ RandAug | 78.8 | 72.94 | 73.54 | 73.76 | 0.9190 | 0.9204 | 0.9302 |
| Deit-tiny | 72.2 | 66.57 | 66.47 | 66.95 | 0.9153 | 0.9197 | 0.9172 |
| Deit-small | 79.9 | 70.65 | 69.72 | 70.07 | 0.9293 | 0.9212 | 0.9277 |

## 4.6 EuroSAT

Helber et al. [21] created EuroSAT from Sentinel-2 satellite images to classify land use and land cover. Past work has improved performance on the dataset through additional training time techniques [45] and using 13 spectral bands [76]. We use RGB images and keep our experimental setup consistent to compare across a range of models. Since there is no set train/test split, we create a 80%/20% class-balanced split.

All models over 60% ImageNet accuracy achieve over 98.5% EuroSAT accuracy, and the majority of our models achieve over 99.0% EuroSAT accuracy. There are certain tasks where using better ImageNet models does not improve performance, and this would be the extreme case where performance saturation is close to being achieved. While it is outside the scope of this study, a next step would be to investigate the remaining errors and find other methods to reduce this last bit of error.

## 5 Additional Studies

### 5.1 Augmentation ablations

In our main experiments, we keep augmentation simple to minimize confounding factors when comparing models. However, it is possible pre-training and fine-tuning with different combinations of augmentations may have different results. This is an important point because different architectures may have different inductive biases and often use different augmentation strategies at pre-training time. To investigate these effects, we run additional experiments on CCT-20 and APTOS to explore the effect of data augmentation on transfer. Specifically, we take ResNet-50 models pre-trained with standard crop and flip augmentation, AugMix [22], and RandAugment [7], and then fine-tune on our default augmentation, AugMix, and RandAugment. We also study DeiT-tiny and Deit-small models by fine-tuning on the same three augmentations mentioned above. We choose to examine DeiT models because they are pre-trained using RandAugment and RandErasing [84]. We increase the number of epochs we fine-tune on from 30 to 50 to account for augmentation. Our experimental results are found in Table 3.

In our ResNet-50 experiments, both AugMix and RandAugment improve performance on ImageNet, but while pre-training with RandAugment improves performance on downstream tasks, pre-training with AugMix does not. Furthermore, fine-tuning with RandAugment usually yields additional performance gains when compared to our default fine-tuning augmentation, no matter which pre-trained model is used. For DeiT models, we found that additional augmentation did not significantly increase performance on the downstream tasks. Thus, as with architectures, augmentation strategies that improve accuracy on ImageNet do not always improve accuracy on real-world tasks.

### 5.2 CLIP models

A natural follow-up to our experiments is to change the source of pre-training data. We examine CLIP models from Radford et al. [52], which use diverse pre-training data and achieve high performance on a variety of downstream datasets. We fine-tune CLIP models on each of our downstream datasets

by linear probing then fine-tuning (LP-FT) [36].[4] Our results are visualized by the purple stars in Appendix H Figure 8. We see that by using a model that takes larger images we can do better than all previous models, and even without the larger images, ViT-L/14 does better on four out of the six datasets. While across all CLIP models the change in pre-training data increases performance for CCT-20, the effect on the other datasets is more complicated. When controlling for architecture changes by only looking at ResNet-50 and ViT/B16, we see that the additional pre-training data helps for CCT-20, HPA, and Cassava, the former two corresponding to the datasets that empirically benefit most from using better ImageNet models. Additional results can be found in Appendix H, while additional fine-tuning details can be found in Appendix I.

## 6 Discussion

**Alternative explanations for saturation.** Whereas Kornblith et al. [31] reported a high degree of correlation between ImageNet and transfer accuracy, we find that better ImageNet models do not consistently transfer better on our real-world tasks. We believe these differences are related to the tasks themselves. Here, we rule out alternative hypotheses for our findings.

Comparison of datasets statistics suggests that the number of classes and dataset size also do not explain the differences from Kornblith et al. [31]. The datasets we study range from two to 28 classes. Although most of the datasets studied in Kornblith et al. [31] have more classes, CIFAR-10 has 10. In Appendix E, we replicate CIFAR-10 results from Kornblith et al. [31] using our experimental setup, finding a strong correlation between ImageNet accuracy and transfer accuracy. Thus, the number of classes is likely not the determining factor. Training set sizes are similar between our study and that of Kornblith et al. [31] and thus also do not seem to play a major role. A third hypothesis is that it is parameter count, rather than ImageNet accuracy, that drives trends. We see that VGG BN models appear to outperform their ImageNet accuracy on multiple datasets, and they are among the largest models by parameter count. However, in Appendix K, we find that model size is also not a good indicator of improved transfer performance on real world datasets.

**Differences between web-scraped datasets and real-world images** We conjecture that it is possible to perform well on most, if not all, web-scraped target datasets simply by collecting a very large amount of data from the Internet and training a very large model on it. Web-scraped target datasets are by definition within the distribution of data collected from the web, and a sufficiently large model can learn that distribution. In support of this conjecture, recent models such as CLIP [52], ALIGN [28], ViT-G [82], BASIC [50], and CoCa [78] are trained on such datasets and achieve high accuracy on many web-scraped benchmarks. But this strategy may not be effective for non-web-scraped datasets, as there is no guarantee that we will train on data that is close in distribution to the target data, even if we train on the entire web. Thus, it makes sense to distinguish these two types of datasets.

There are clear differences in image distribution between the non-web-scraped datasets we consider and web-scraped datasets considered by previous work. In Figure 3 and Appendix L, we compute Fréchet inception distance (FID) [23] between ImageNet and each of the datasets we study in this work as well as the ones found in Kornblith et al. [31]. The real-world datasets are further from ImageNet than datasets in Kornblith et al. [31], implying that there is a large amount of distribution shift between web-scraped datasets and real-world datasets. However, FID is only a proxy measure and may not capture all factors that lead to differences in transferability.

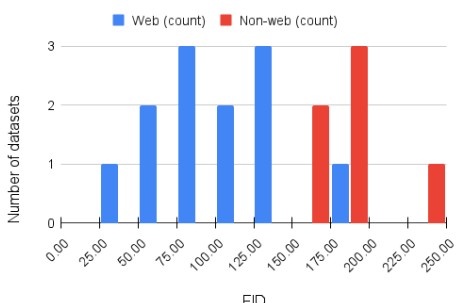

Figure 3: FID scores vs ImageNet for the datasets we study in this work (red), and the web-scraped datasets studied by Kornblith et al. [31] (blue).

Whereas web-scraped data is cheap to acquire, real-world data can be more expensive. Ideally, progress in computer vision architectures should improve performance not just on web-scraped data, but also on real-world tasks. Our results suggest that the latter has not happened. Gains in ImageNet accuracy over the last decade have primarily come from improving and scaling architectures, and past work has shown that these gains generally transfer to other web-scraped datasets, regardless of size [63, 31, 41, 73, 30]. However, we find that improvements

---

[4]We use LP-FT because, in past experiments, we have found that LP-FT makes hyperparameter tuning easier for CLIP models, but does not significantly alter performance when using optimal hyperparameters.

arising from architecture generally do not transfer to non-web-scraped tasks. Nonetheless, data augmentation and other tweaks can provide further gains on these tasks.

**Recommendations towards better benchmarking.** While it is unclear whether researchers have over-optimized for ImageNet, our work suggests that researchers should explicitly search for methods that improve accuracy on real-world non-web-scraped datasets, rather than assuming that methods that improve accuracy on ImageNet will provide meaningful improvements on real-world datasets as well. Just as there are methods that improve accuracy on ImageNet but not on the tasks we investigate, there may be methods that improve accuracy on our tasks but not ImageNet. The Kaggle community provides some evidence for the existence of such methods; Kaggle submissions often explore architectural improvements that are less common in traditional ImageNet pre-trained models. To measure such improvements on real-world problems, we suggest simply using the average accuracy across our tasks as a benchmark for future representation learning research.

Further analysis of our results shows consistencies in the accuracies of different models across the non-web-scraped datasets, suggesting that accuracy improvements on these datasets may translate to other datasets. For each dataset, we use linear regression to predict model accuracies on the target dataset as a linear combination of ImageNet accuracy and accuracy averaged across the other real-world datasets. We perform an F-test to determine whether the average accuracy on other real-world datasets explains significant variance beyond that explained by ImageNet accuracy. We find that this F-test is significant on all datasets except EuroSAT, where accuracy may be very close to ceiling (see further analysis in Appendix M.1). Additionally, in Appendix M.2 we compare the Spearman rank correlation (i.e., the Pearson correlation between ranks) between each dataset and the accuracy averaged across the other real-world datasets to the Spearman correlation between each dataset and ImageNet. We find that the correlation with the average over real-world datasets is higher than the correlation with ImageNet and statistically significant for CCT-20, APTOS, HPA, and Cassava. Thus, there is some signal in the average accuracy across the datasets that we investigate that is not captured by ImageNet top-1 accuracy.

Where do our findings leave ImageNet? We suspect that most of the methodological innovations that help on ImageNet are useful for some real-world tasks, and in that sense it has been a successful benchmark. However, the innovations that improve performance on industrial web-scraped datasets such as JFT [63] or IG-3.5B-17k [41] (e.g., model scaling) may be almost entirely disjoint from the innovations that help with the non-web-scraped real-world tasks studied here (e.g., data augmentation strategies). We hope that future benchmarks will include more diverse datasets to encourage a more comprehensive approach to improving learning algorithms.

## Acknowledgements

We would like to thank Samuel Ainsworth, Sara Beery, Gabriel Ilharco, Pieter-Jan Kindermans, Sarah Pratt, Matthew Wallingford, Ross Wightman, and Mitchell Wortsman for valuable conversations while working on this project. We would especially like to thank Sarah Pratt for help with early experimentation and brainstorming.

We would also like to thank Hyak computing cluster at the University of Washington and the Google TPU Research Cloud program for access to compute resources that allowed us to run our experiments.

This work is in part supported by the NSF AI Institute for Foundations of Machine Learning (IFML), Open Philanthropy, Google, and the Allen Institute for AI.

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

# Appendix

## A Detailed experiment results

Table 4: For each ImageNet pre-trained model, we provide the best performing model when fine-tuned on each dataset across our hyperparameter grid

| Model | ImageNet top-1 | CCT20 | APTOS | HPA | Melanoma | Cassava | EuroSAT |
|---|---|---|---|---|---|---|---|
| AlexNet | 56.5 | 63.59 | 0.8835 | 0.3846 | 0.9283 | 82.58 | 97.93 |
| SqueezeNet 1.1 | 58.2 | 66.36 | 0.9021 | 0.3972 | 0.9073 | 85.15 | 98.07 |
| ShuffleNetV2x0.5 | 60.6 | 66.37 | 0.9227 | 0.5867 | 0.9289 | 85.64 | 98.56 |
| MobileNet V3 small | 67.7 | 66.01 | 0.9230 | 0.6108 | 0.9455 | 85.81 | 99.15 |
| ShuffleNetV2x1.0 | 69.4 | 69.27 | 0.9202 | 0.6202 | 0.9418 | 87.33 | 98.91 |
| VGG-13 BN | 71.6 | 75.06 | 0.9268 | 0.6794 | 0.9529 | 88.99 | 98.85 |
| DeiT-tiny | 72.2 | 68.77 | 0.9130 | 0.5777 | 0.9510 | 86.25 | 99.11 |
| VGG-16 BN | 73.4 | 75.93 | 0.9287 | 0.6791 | 0.9531 | 88.45 | 98.93 |
| DenseNet-121 | 74.4 | 74.66 | 0.9287 | 0.7019 | 0.9514 | 87.80 | 99.06 |
| ResNet-50 | 76.1 | 73.96 | 0.9215 | 0.6718 | 0.9524 | 87.75 | 99.19 |
| ResNeXt-50-32x4d | 77.6 | 73.73 | 0.9212 | 0.6906 | 0.9588 | 88.15 | 99.24 |
| EfficientNet B0 | 77.7 | 71.02 | 0.9195 | 0.6942 | 0.9456 | 87.63 | 98.80 |
| ResNet-152 | 78.3 | 74.05 | 0.9228 | 0.6732 | 0.9562 | 87.75 | 99.15 |
| ViT-B/16 | 78.7 | 72.07 | 0.9262 | 0.5852 | 0.9600 | 86.63 | 99.28 |
| DeiT-small | 79.9 | 71.41 | 0.9205 | 0.6148 | 0.9583 | 87.19 | 99.20 |
| Inception-ResNet v2 | 80.4 | 70.68 | 0.9168 | 0.6882 | 0.9483 | 87.84 | 98.93 |
| ConvNext-tiny | 82.5 | 78.51 | 0.9297 | 0.6992 | 0.9628 | 88.89 | 99.11 |
| PNASNet-5 large | 82.9 | 75.21 | 0.9271 | 0.6941 | 0.9584 | 87.77 | 99.17 |
| EfficientNet B4 | 83.4 | 73.49 | 0.9211 | 0.6954 | 0.9552 | 88.36 | 98.70 |

See the following link for experiment results across hyperparameters: `https://docs.google.com/spreadsheets/d/1aDeuTHOV1Kid_JMRUt3sF1N76LUCAMDQOO7Ykjo3Z4U/edit?usp=sharing`.

# B   Main figure variations

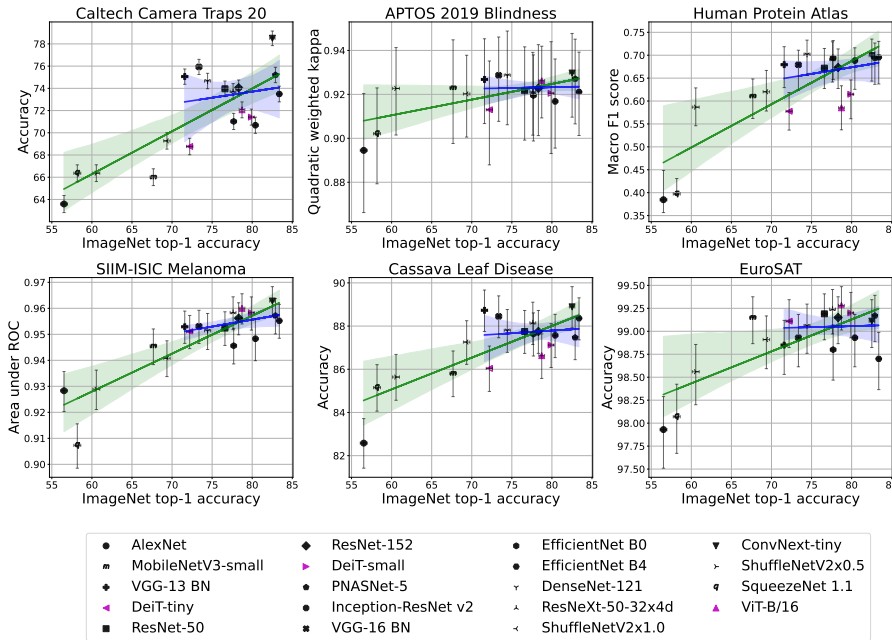

Figure 4: Figure 1 with error bars. Green is linear trend of all models, while blue is linear trend for models above 70% ImageNet accuracy. We use 95% confidence intervals computed with Clopper-Pearson for accuracy metrics and bootstrap with 10,000 trials for other metrics.

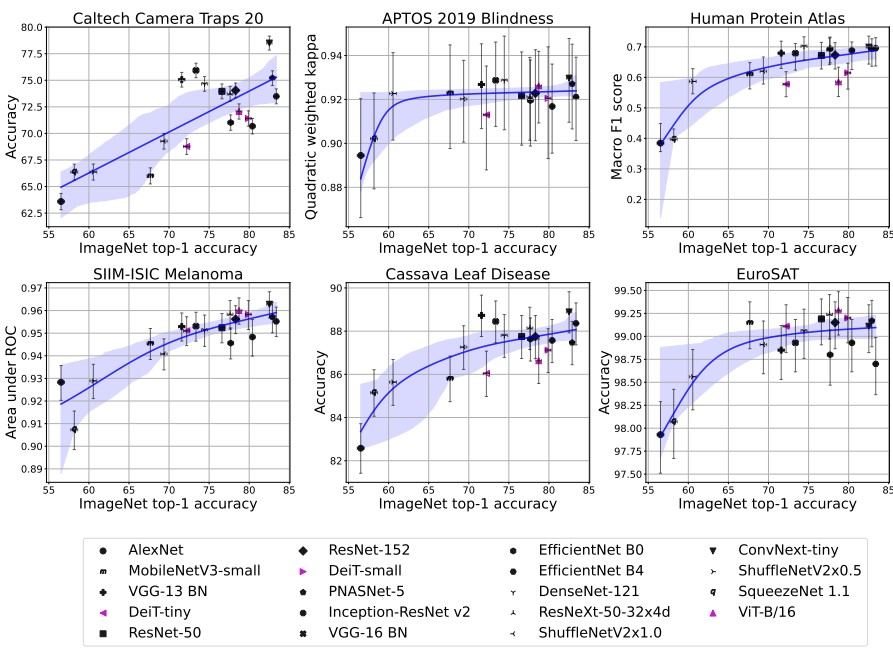

Figure 5: Figure 4 with spline interpolation fits instead of linear fits.

## C  Experiment setup

### C.1  Models

Table 5: We examine the effectiveness of transfer learning from a number of models pretrained on ImageNet, including both CNNs and Vision Transformers.

| Model | ImageNet top-1 | # params | Year Released |
|---|---|---|---|
| AlexNet [35] | 56.5 | 61M | 2012 |
| SqueezeNet 1.1 [27] | 58.2 | 1.2M | 2016 |
| ShuffleNetV2x0.5 [40] | 60.6 | 1.4M | 2018 |
| MobileNet V3 small [24] | 67.7 | 2.5M | 2019 |
| ShuffleNetV2x1.0 [40] | 69.4 | 2.3M | 2018 |
| VGG-13 BN [61] | 71.6 | 133M | 2014/2015 |
| DeiT-tiny [69] | 72.2 | 5.7M | 2020 |
| VGG-16 BN [61] | 73.4 | 138M | 2014/2015 |
| DenseNet-121 [25] | 74.4 | 8.0M | 2016 |
| ResNet-50 [20] | 76.1 | 26M | 2015 |
| ResNeXt-50-32x4d [74] | 77.6 | 25M | 2016 |
| EfficientNet B0 [66] | 77.7 | 5.3M | 2019 |
| ResNet-152 [20] | 78.3 | 60M | 2015 |
| ViT-B/16 [11, 62] | 78.7 | 304M | 2020 |
| DeiT-small [69] | 79.9 | 22M | 2020 |
| Inception-ResNet v2 [64] | 80.4 | 56M | 2016 |
| ConvNext-tiny [39] | 82.5 | 29M | 2022 |
| PNASNet-5 large [37] | 82.9 | 86M | 2017 |
| EfficientNet B4 [66] | 83.4 | 19M | 2019 |

We examine 19 model architectures in this work that cover a diverse range of accuracies on ImageNet in order to observe the relationship between ImageNet performance and target dataset performance. In addition to the commonly used CNNs, we also include data-efficient image transformers (DeiT) due to the recent increase in usage of Vision Transformers. Additional model details are in Table 5.

### C.2  Hyperparameter Grid

Hyperparameter tuning is a key part of neural network training, as using suboptimal hyperparameters can lead to suboptimal performance. Furthermore, the correct hyperparameters vary across both models and training data. To get the best performance out of each model, we train each model on AdamW with a cosine decay learning rate schedule, SGD with a cosine decay learning rate schedule, and SGD with a multi-step decay learning rate schedule. We also grid search for optimal initial learning rate and weight decay combinations, searching logarithmically between $10^{-1}$ to $10^{-4}$ for SGD learning rate, $10^{-2}$ to $10^{-5}$ for AdamW learning rate, and $10^{-3}$ to $10^{-6}$ as well as 0 for weight decay. All models are pretrained on ImageNet and then fine-tuned on the downstream task. Additional training details for each dataset can be found in Appendix D. We also run our hyperparameter grid on CIFAR-10 in Appendix E to verify that we find a strong relationship between ImageNet and CIFAR-10 accuracy as previously reported by Kornblith et al. [31].

## D  Training details by dataset (ImageNet models)

Experiments on Cassava Leaf Disease, SIIM-ISIC Melanoma, and EuroSAT datasets were ran on TPU v2-8s, while all other datasets were ran on NVIDIA A40s.

All experiments were ran with mini-batch size of 128.

For SGD experiments, we use Nesterov momentum, set momentum to 0.9, and try learning rates of 1e-1, 1e-2, 1e-3, and 1e-4. For AdamW experiments, we try learning rates of 1e-2, 1e-3, 1e-4, 1e-5. For all experiments, we try weight decays of 1e-3, 1e-4, 1e-5, 1e-6, and 0.

For all experiments, we use weights that are pretrained on ImageNet. AlexNet, DenseNet, MobileNet, ResNet, ResNext, ShuffleNet, SqueezeNet and VGG models are from torchvision, while ConvNext, DeiT, EfficientNet, InceptionResNet, and PNASNet models are from timm. Additionally, we normalize images to ImageNet's mean and standard deviation.

For EuroSAT we random resize crop to 224 with area at least 0.65.

For all other datasets, we random resize crop with area at least 0.65 to 224 for DeiT models, and 256 for all other models. Additionally, we use horizontal flips. For Human Protein Atlas, Cassava Leaf Disease, and SIIM-ISIC Melanoma, we also use vertical flips.

For SIIM-ISIC Melanoma, we train for 10 epochs, and for the step scheduler decay with factor 0.1 at 5 epochs.

For all other datasets, we train for 30 epochs, and for the step scheduler decay with factor 0.1 at 15, 20, and 25 epochs.

# E CIFAR-10 on hyperparameter grid

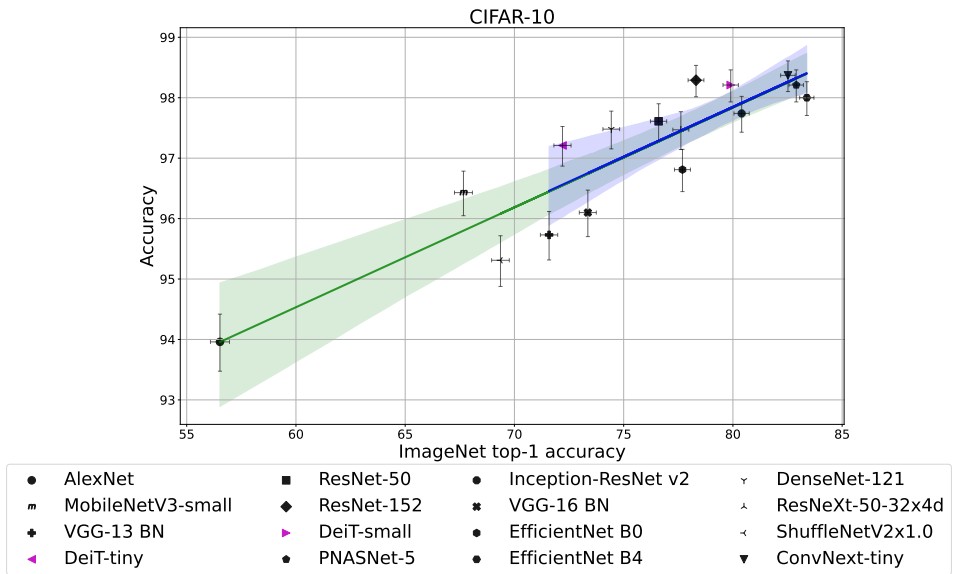

Figure 6: Transfer performance across models from ImageNet to CIFAR-10. Green linear trend is computed across all models, while blue linear trend is restricted to models above 70% ImageNet accuracy. We use 95% confidence intervals computed with Clopper-Pearson.

# F APTOS 2019 Blindness Detection ablations

Scores presented are submissions to the Kaggle leaderboard. All scores are evaluated with quadratic weighted kappa. Within each entry, we first present the private leaderboard score, then the public leaderboard score. The private leaderboard represents 85% of the test data, while the public leaderboard is the remaining 15%.

Models used here are trained using AdamW with a cosine scheduler. We random resize crop to 512, use random rotations, and use color jitter (brightness=0.2, contrast=0.2, saturation=0.2, hue=0.1). We train on all the available training data, no longer using the local train/validation split mentioned in the main text. This includes both the training data in the 2019 competition, as well as data from a prior 2015 diabetic retinopathy competition.

Table 6: Comparing various models with additional interventions by evaluating on the Kaggle leaderboard.

| | lr \wd | 1.00E-04 | 1.00E-05 | 1.00E-06 |
|---|---|---|---|---|
| ResNet-50 | 1.00E-03 | 0.8610 / 0.6317 | 0.8570 / 0.6180 | 0.8548 / 0.6646 |
| | 1.00E-04 | 0.8952 / 0.7531 | 0.8918 / 0.7204 | **0.8961 / 0.7547** |
| ResNet-152 | 1.00E-03 | 0.8658 / 0.6812 | 0.8686 / 0.6612 | 0.8640 / 0.6554 |
| | 1.00E-04 | **0.8898 / 0.7164** | 0.8836 / 0.6946 | 0.8859 / 0.6947 |
| Inception-Resnet-v2 | 1.00E-03 | 0.8933 / 0.7748 | 0.8905 / 0.7565 | **0.8960 / 0.7585** |
| | 1.00E-04 | 0.8897 / 0.7210 | 0.8929 / 0.7420 | 0.8944 / 0.7439 |

Table 7: Comparing the effect of augmentation on Kaggle leaderboard scores. More augmentation is as described earlier in this section. Less augmentation only uses random resize crop with at least 0.65 area and horizontal flips.

| | lr \wd | 1.00E-04 | 1.00E-05 | 1.00E-06 |
|---|---|---|---|---|
| ResNet-50 less aug | 1.00E-03 | **0.8669 / 0.6405** | 0.8520 / 0.6013 | 0.8613 / 0.6269 |
| | 1.00E-04 | 0.8525 / 0.6115 | 0.8570 / 0.6431 | 0.8483 / 0.6147 |
| | 1.00E-05 | 0.8186 / 0.5071 | 0.8287 / 0.5647 | 0.8288 / 0.5328 |
| ResNet-50 more aug | 1.00E-03 | 0.8440 / 0.6432 | 0.8547 / 0.6856 | 0.8524 / 0.7125 |
| | 1.00E-04 | 0.8948 / 0.7490 | 0.8972 / 0.7693 | **0.8999 / 0.7758** |
| | 1.00E-05 | 0.8724 / 0.7370 | 0.8685 / 0.7567 | 0.8623 / 0.7376 |

## G  Melanoma Metric Comparison

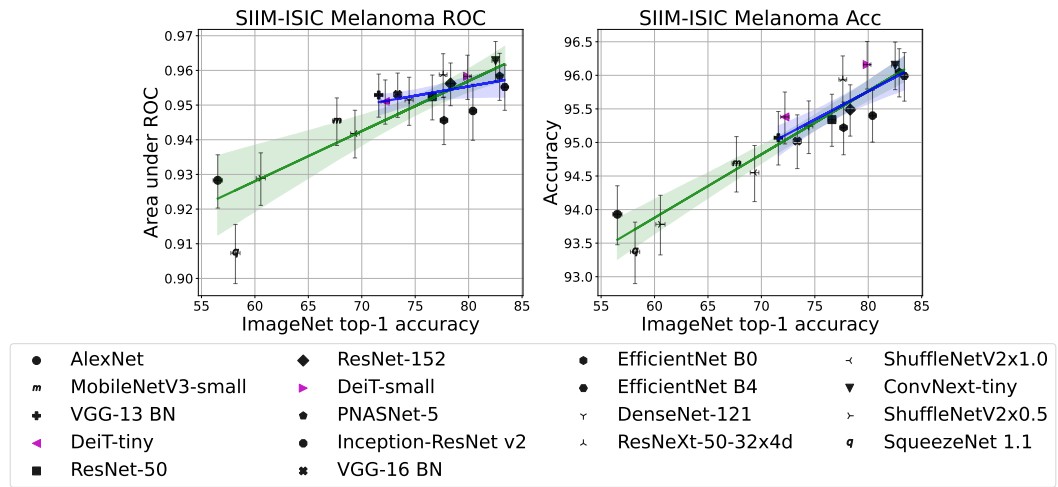

| | | | | | | | |
|---|---|---|---|---|---|---|---|
| ● | AlexNet | ◆ | ResNet-152 | ● | EfficientNet B0 | ◁ | ShuffleNetV2x1.0 |
| m | MobileNetV3-small | ▶ | DeiT-small | ● | EfficientNet B4 | ▼ | ConvNext-tiny |
| ✦ | VGG-13 BN | ● | PNASNet-5 | ⊤ | DenseNet-121 | ➤ | ShuffleNetV2x0.5 |
| ◀ | DeiT-tiny | ● | Inception-ResNet v2 | ⊥ | ResNeXt-50-32x4d | q | SqueezeNet 1.1 |
| ■ | ResNet-50 | ✳ | VGG-16 BN | | | | |

Figure 7: Comparing transfer performance from ImageNet to Melanoma when using different metrics. Green linear trend is computed across all models, while blue linear trend is restricted to models above 70% ImageNet accuracy. Using accuracy implies that better ImageNet models transfer better; however, ROC is a better metric for this task.

# H    CLIP experiment details

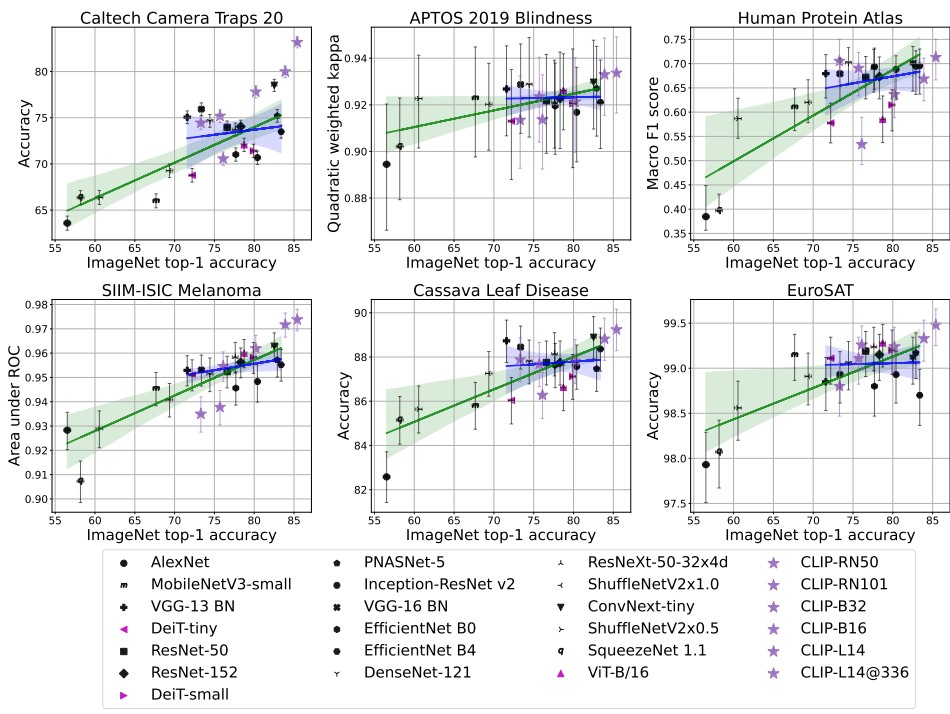

Figure 8:  Figure 4 with CLIP models overlaid (purple stars).  The best CLIP models do better than all the ImageNet models, but when looking across all CLIP models, the patterns are more complicated.

Table 8:  For each CLIP pre-trained model, we provide the best performing model when fine-tuned on each dataset across our LP-FT hyperparameter grid

| Model | ImageNet top-1 | CCT20 | APTOS | HPA | Melanoma | Cassava | EuroSAT |
|---|---|---|---|---|---|---|---|
| CLIP-RN50 | 73.3 | 74.45 | 0.9135 | 0.7053 | 0.9350 | 87.89 | 98.80 |
| CLIP-RN101 | 75.7 | 75.19 | 0.9235 | 0.6909 | 0.9378 | 87.68 | 99.11 |
| CLIP-B32 | 76.1 | 70.57 | 0.9137 | 0.5338 | 0.9546 | 86.28 | 99.26 |
| CLIP-B16 | 80.2 | 77.81 | 0.9213 | 0.6365 | 0.9619 | 87.82 | 99.24 |
| CLIP-L14 | 83.9 | 79.99 | 0.9330 | 0.6687 | 0.9717 | 88.82 | 99.33 |
| CLIP-L14@336 | 85.4 | 83.17 | 0.9337 | 0.7131 | 0.9738 | 89.24 | 99.48 |

Table 9:  We directly compare models pre-trained on ImageNet with models pre-trained on OpenAI's CLIP data. Specifically, we look at ResNet 50 and ViT B/16.

| Model | ImageNet top-1 | CCT20 | APTOS | HPA | Melanoma | Cassava | EuroSAT |
|---|---|---|---|---|---|---|---|
| IN-ResNet-50 | 76.1 | 73.96 | 0.9215 | 0.6718 | 0.9524 | 87.75 | 99.19 |
| CLIP-RN50 | 73.3 | 74.45 | 0.9135 | 0.7053 | 0.9350 | 87.89 | 98.80 |
| IN-ViT-B/16 | 78.7 | 72.07 | 0.9262 | 0.5852 | 0.9600 | 86.63 | 99.28 |
| CLIP-B16 | 80.2 | 77.81 | 0.9213 | 0.6365 | 0.9619 | 87.82 | 99.24 |

# I CLIP fine-tuning details

We fine-tune by running a linear probe, followed by end-to-end fine-tuning on the best model from the first part. We keep total epochs consistent with the previous models, with a third of the epochs going toward linear probing. We use AdamW with a cosine decay schedule. During the linear probe, we search over $10^{-1}$, $10^{-2}$, and $10^{-3}$ learning rates, and during fine-tuning, we search over $10^{-4}$, $10^{-5}$, and $10^{-6}$ learning rates. For both parts, we search over $10^{-3}$ to $10^{-6}$ and 0 for weight decay.

# J Creation information for datasets studied in Kornblith et al. [31]

Table 10: We find that the 12 datasets studied in Kornblith et al. [31] come from web scraping.

| Dataset | Origin | Additional information |
|---|---|---|
| Food-101 | foodspotting.com | Users upload an image of their food and annotate the type of food; categories chosen by popularity |
| CIFAR-10 | TinyImages | Web crawl |
| CIFAR-100 | TinyImages | Web crawl |
| Birdsnap | Flickr | Also used MTurk |
| SUN397 | Web search engines | Also used WordNet |
| Stanford Cars | Flickr, Google, Bing | Also used MTurk |
| FGVC Aircraft | airliners.net | Images taken by 10 photographers |
| Pascal VOC 2007 Cls. | Flickr | N/A |
| Describable Textures | Google and Flickr | Also used MTurk |
| Oxford-IIT Pets | Flickr, Google, Catster, Dogster | Catster and Dogster are social websites for collecting and discussing pet images |
| Caltech-101 | Google | 97 categories chosen from Webster Collegiate Dictionary categories associated with a drawing |
| Oxford 102 Flowers | Mostly collected from web | A small number of images acquired by the paper authors taking the pictures |

# K    Relationship between model size and transfer performance

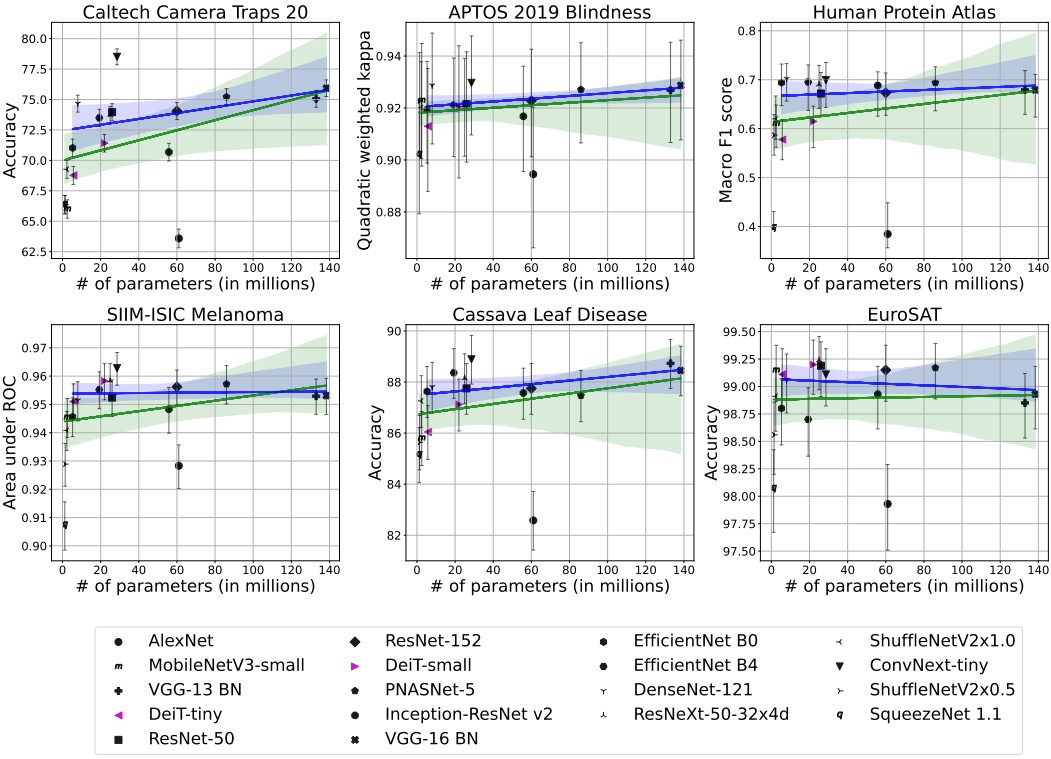

Figure 9: We compare model size with downstream transfer performance. Again we use separate trend lines for all models (green) and only those above 70% ImageNet accuracy (blue). We use 95% confidence intervals computed with Clopper-Pearson for accuracy metrics and bootstrap with 10,000 trials for other metrics.

## L    FID score details

Table 11: We calculate FID scores between the ImageNet validation set and each of the datasets we study, as well as between the ImageNet validation set and each of the datasets in Kornblith et al. [31]. We found that dataset size affects FID score, so we take a 3,662 subset of each downstream dataset. Note that 3,662 is the size of APTOS, which is the smallest dataset.

| Dataset | FID |
|---|---|
| CCT-20 | 162.69 |
| APTOS | 196.24 |
| HPA | 230.70 |
| Cassava | 179.24 |
| Melanoma | 186.34 |
| EuroSAT | 151.85 |
| Food-101 | 108.35 |
| CIFAR-10 | 132.53 |
| CIFAR-100 | 120.72 |
| Birdsnap | 94.08 |
| SUN397 | 62.95 |
| Stanford Cars | 143.35 |
| FGVC Aircraft | 183.35 |
| Pascal VOC 2007 Cls. | 39.84 |
| Describable Textures | 89.13 |
| Oxford-IIT Pets | 77.27 |
| Caltech-101 | 50.77 |
| Oxford 102 Flowers | 140.21 |

## M    Predictive power of accuracy on non-web-scraped datasets on novel datasets

We observe that, on many non-web-scraped datasets, accuracy correlates only weakly with ImageNet accuracy. It is thus worth asking whether other predictors might correlate better. In this section, we examine the extent to which accuracy on a given non-web-scraped target dataset can be predicted from the accuracy on the other non-web-scraped target datasets.

### M.1    F-test

We can further measure the extent to which the averages of the five other datasets *beyond* the predictive power provided by ImageNet by using F-tests. For each target task, we fit a linear regression model that predicts accuracy as either ImageNet accuracy or the average accuracy on the other five non-web-scraped datasets, and a second linear regression model that predicts accuracy as a function of both ImageNet accuracy and the average accuracy on the other five datasets. Since the first model is nested within the second, the second model must explain at least as much variance as the first. The F-test measures whether the increase in explained variance is significant. For these experiments, we logit-transform accuracy values and standardize them to zero mean and unit variance before computing the averages, as in the middle column of Table 13.

Results are shown in Table 12. The average accuracy across the other five datasets explains variance beyond that explained by ImageNet accuracy alone on five of the six datasets. The only exception is EuroSAT, where the range of accuracies is low (most models get ~99%) and a significant fraction of the variance among models may correspond to noise. By contrast, ImageNet accuracy explains variance beyond the average accuracy only on two datasets (APTOS and Melanoma). These results indicate that there are patterns in how well different models transfer to non-web-scraped data that are not captured by ImageNet accuracy alone, but are captured by the accuracy on other non-web-scraped datasets.

Table 12: Results of the F-test described in Section M.1. "+Avg. across datasets" tests whether a model that includes both ImageNet accuracy and the average accuracy across the 5 other datasets explains more variance than a model that includes only ImageNet accuracy. "+ImageNet" tests whether a model that includes both predictors explains more variance than a model that includes only the average accuracy across the 5 other datasets. In addition to $F$ and $p$ values, we report adjusted $R^2$ for all models. $p$-values $< 0.05$ are bold-faced.

| | +Avg. across datasets | | +ImageNet | | Adj. $R^2$ | Adj. $R^2$ | Adj. $R^2$ |
|---|---|---|---|---|---|---|---|
| Dataset | $F(1, 16)$ | $p$-value | $F(1, 16)$ | $p$-value | (ImageNet-only) | (Average-only) | (Both predictors) |
| CCT-20 | 8.2 | **0.01** | 0.69 | 0.42 | 0.56 | 0.70 | 0.69 |
| APTOS | 31.0 | **0.00004** | 4.6 | **0.047** | 0.34 | 0.71 | 0.76 |
| HPA | 11.8 | **0.003** | 0.84 | 0.37 | 0.60 | 0.76 | 0.76 |
| Melanoma | 5.8 | **0.03** | 7.8 | **0.01** | 0.74 | 0.71 | 0.79 |
| Cassava | 13.2 | **0.002** | 0.14 | 0.71 | 0.55 | 0.75 | 0.74 |
| EuroSAT | 2.9 | 0.11 | 0.72 | 0.41 | 0.43 | 0.52 | 0.49 |

## M.2   Spearman correlation

Table 13: We measure the Spearman correlation between each dataset with either the average of the 5 other datasets we study, or with ImageNet. Normalization is done by logit transforming accuracies, and then standardizing to zero mean and unit variance. The results suggest that using additional datasets is more predictive of model performance than just using ImageNet.

| | Avg of 5 others (unnormalized) | | Avg of 5 others (normalized) | | ImageNet | |
|---|---|---|---|---|---|---|
| Dataset | $\rho$ | $p$-value | $\rho$ | $p$-value | $\rho$ | $p$-value |
| CCT-20 | 0.8684 | 0.0000 | 0.9263 | 0.0000 | 0.5825 | 0.0089 |
| APTOS | 0.7205 | 0.0005 | 0.6950 | 0.0010 | 0.3010 | 0.2105 |
| HPA | 0.7351 | 0.0003 | 0.6825 | 0.0013 | 0.6491 | 0.0026 |
| Melanoma | 0.6561 | 0.0023 | 0.7807 | 0.0000 | 0.7667 | 0.0001 |
| Cassava | 0.8872 | 0.0000 | 0.7442 | 0.0003 | 0.5222 | 0.0218 |
| EuroSAT | 0.3030 | 0.2073 | 0.3821 | 0.1065 | 0.4734 | 0.0406 |

# N  Pre-training augmentation details

Table 14:  For each ImageNet pre-trained model, we provide the augmentation strategy used during pre-training time.

| Model | Augmentation |
|---|---|
| AlexNet | Resize + Crop + Flip |
| SqueezeNet 1.1 | Resize + Crop + Flip |
| ShuffleNetV2x0.5 | AutoAugment (TrivialAugmentWide) + RandErasing + MixUp + CutMix |
| MobileNet V3 small | AutoAugment (ImageNet/Default)+ RandErasing |
| ShuffleNetV2x1.0 | AutoAugment (TrivialAugmentWide) + RandErasing + MixUp + CutMix |
| VGG-13 BN | Resize + Crop + Flip |
| DeiT-tiny | RandAugment + RandErasing |
| VGG-16 BN | Resize + Crop + Flip |
| DenseNet-121 | Resize + Crop + Flip |
| ResNet-50 | Resize + Crop + Flip |
| ResNeXt-50-32x4d | Resize + Crop + Flip |
| EfficientNet B0 | RandAugment |
| ResNet-152 | Resize + Crop + Flip |
| ViT-B/16 | RandAugment + MixUp |
| DeiT-small | RandAugment + RandErasing |
| Inception-ResNet v2 | Inception Preprocessing (Color Distort + Resize + Crop + Flip) |
| ConvNext-tiny | AutoAugment (TrivialAugmentWide) + RandErasing + MixUp + CutMix |
| PNASNet-5 large | Whiten + Resize + Crop + Flip |
| EfficientNet B4 | RandAugment |

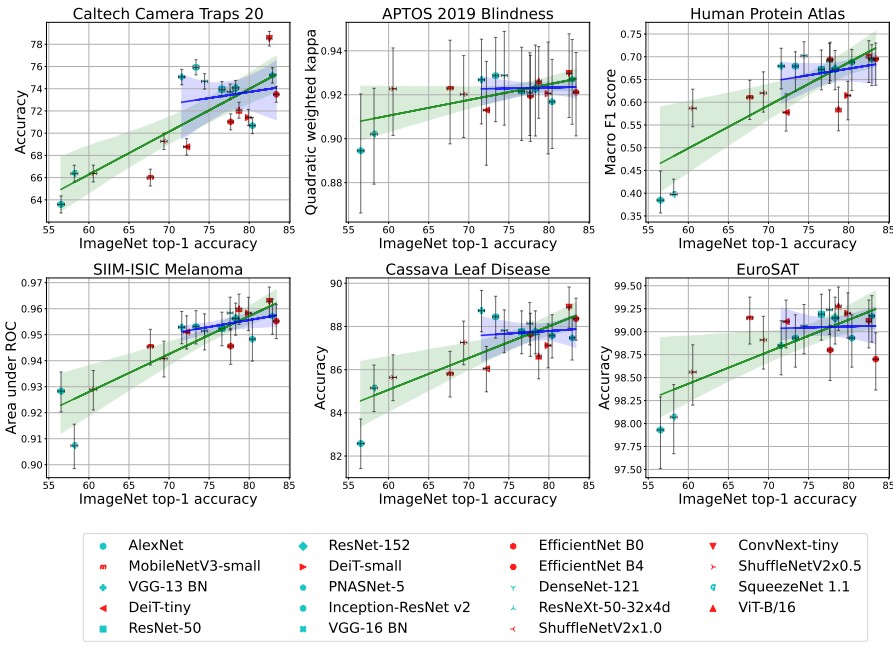

Figure 10:  Figure 1 with points colored by general pre-training augmentation strategy. Cyan points use simple augmentation (resize, crops, flips, etc.), and red points use automatic augmentation (RandAugment, AutoAugment, TrivialAugmentWide).

# O  Additional Models

Here we include four additional low accuracy models and two newer high accuracy models that were not originally included in our study. While the inclusion of the newer models causes the linear fit slopes to go up slightly, progress on downstream tasks still significantly lags behind progress on ImageNet for some of the datasets.

## O.1  Models

| Model | ImageNet top-1 | # params | Year Released |
|-------|---------------|----------|---------------|
| TinyNet e [18] | 59.9 | 2.0M | 2020 |
| DLA46 c [77] | 64.9 | 1.3M | 2018 |
| DLA46x c [77] | 66.0 | 1.1M | 2018 |
| TinyNet d [18] | 67.0 | 2.3M | 2020 |
| SWIN Base [38] | 83.6 | 88M | 2021 |
| CAFormer B36 [79] | 85.5 | 99M | 2022 |

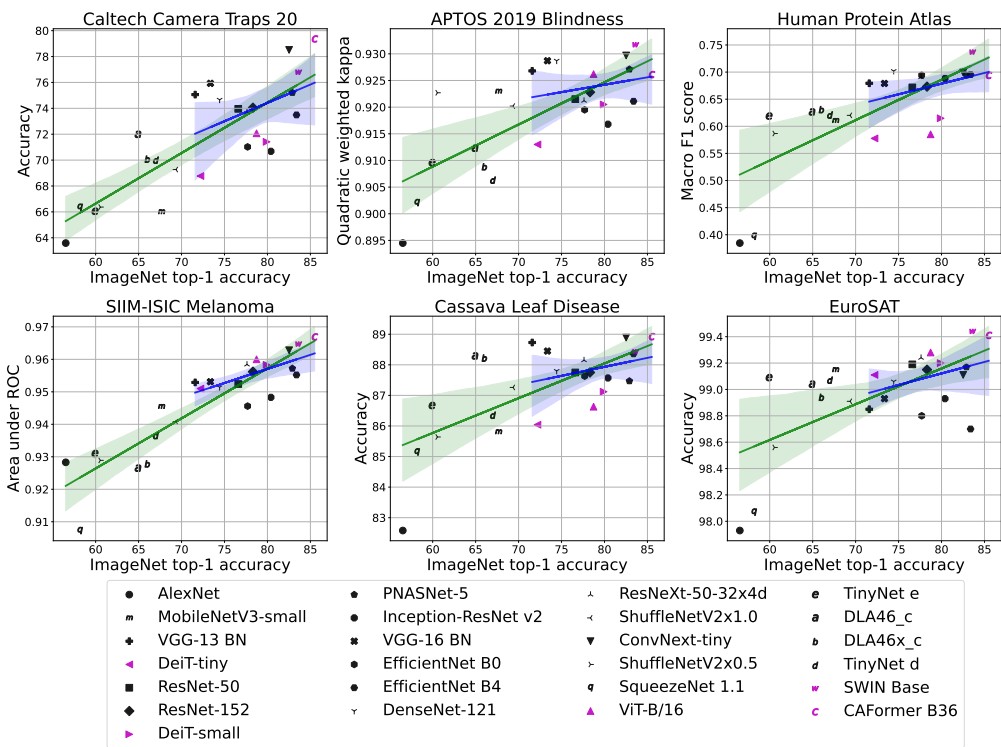

Figure 11: Figure 1 with additional models. Green is linear trend of all models, while blue is linear trend for models above 70% ImageNet accuracy.

Table 15: We summarize the blue regression lines from Figure 11, calculated on models above 70% ImageNet accuracy, with their correlation and slope. Slope is calculated so that all metrics have a range from 0 to 100.

| Dataset | Correlation | Slope |
|---|---|---|
| Caltech Camera Traps | 0.43 | 0.28 |
| APTOS 2019 Blindness | 0.23 | 0.03 |
| Human Protein Atlas | 0.39 | 0.39 |
| SIIM-ISIC Melanoma | 0.63 | 0.08 |
| Cassava Leaf Disease | 0.31 | 0.06 |
| EuroSAT | 0.36 | 0.02 |

Table 16: For each ImageNet pre-trained model, we provide the best performing model when fine-tuned on each dataset across our hyperparameter grid

| Model | ImageNet top-1 | CCT20 | APTOS | HPA | Melanoma | Cassava | EuroSAT |
|---|---|---|---|---|---|---|---|
| TinyNet e | 59.9 | 66.05 | 0.9096 | 0.6189 | 0.9310 | 86.67 | 99.09 |
| DLA46 c | 64.9 | 71.99 | 0.9123 | 0.6265 | 0.9265 | 88.29 | 99.04 |
| DLA46x c | 66.0 | 70.07 | 0.9088 | 0.6306 | 0.9278 | 88.22 | 98.94 |
| TinyNet d | 67.0 | 69.99 | 0.9063 | 0.6208 | 0.9365 | 86.35 | 99.07 |
| SWIN Base | 83.6 | 76.79 | 0.9317 | 0.7369 | 0.9647 | 88.43 | 99.44 |
| CAFormer B36 | 85.5 | 79.36 | 0.9261 | 0.6939 | 0.9669 | 88.92 | 99.41 |

