# OpenReview forum: "Does progress on ImageNet transfer to real-world datasets?"
_NeurIPS.cc/2023/Track/Datasets_and_Benchmarks — NeurIPS 2023 Datasets and Benchmarks Poster_

### Official Review · Reviewer_ok7Z · 2023-07-19
**Does progress on ImageNet transfer to real-world datasets?**

**Rating:** 4
**Confidence:** 5

**Strengths:**

- Studies an interesting and highly relevant question: Does progress on ImageNet transfer to real-world datasets?
- The paper is clearly written and easy to follow
- There is a thorough (given the space) report on previous work
- Experiments are performed on multiple diverse models and downstream datasets

**Additional Feedback:**

- I understand that the D&B track allowed to choose authors between single-/double-blind. However, the reason for this is that maintaining anonymity with (large) datasets is hard. In this case, the paper could have also been posted anonymously. As some of the authors are well-known, I'd recommend submitting anonymously in the future if the authors are interested in an unbiased evaluation of their work.

**Clarity:**

The paper is generally clearly written. I would still propose moving some experimental setup details from the appendix to the main paper to clearly emphasize that the authors study end-to-end fine-tuning. The same applies to Sec. 4.7.

**Correctness:**

The claims made in the submission appear to be correct within the scope of the aforementioned limitations. One deviation of this might be the claims about "web-scraped datasets" (see above).

**Documentation:**

There is sufficient detail to reproduce the details and the code is available at https://github.com/mlfoundations/imagenet-applications-transfer

**Ethics:**

No concerns

**Limitations:**

The authors discuss some limitations of their work in the discussion section. However, I think it would be worth mentioning, that they only investigate end-to-end finetuning. It is thus unclear if other techniques (e.g. only fine-tuning the last layer, domain adaption, etc.) would show similar results. Further, the authors only explore 6 downstream datasets. It remains to be shown whether these are a good proxy for other datasets.

I do not see any potential negative societal impact.

**Opportunities For Improvement:**

- The authors aim to specifically investigate the role of neural network architectures (L29). However, when evaluated in the fine-tuning setting, this also means that the authors naturally include all the pertaining biases. From what I understand, the authors rely on provided checkpoints that may have been trained with different hyperparameters such as optimizers, augmentations, schedulers etc. - as such I am not sure how well this can be decoupled from the architecture especially when comparing "legacy" models such as AlexNet to modern and highly-optimized models such as ViT.
- The authors repeatedly differentiate between web-scraped datasets (ImageNet, CIFAR, Pascal VOC) and the ones studied here (e.g. carefully curated). They repeatedly state that there are differences in transfer between web-scraped datasets and the ones in this study. While the numbers definitely prove this, I am concerned that this may be due to the semantic similarity to ImagNet rather than some aspect of "web-scraping". Contrarily, the downstream datasets do not share any similarities with ImageNet and are thus OOD. Also, the examined datasets are publicly hosted by Kaggle and others, so anyone could crawl them now and build a new dataset. While the resulting dataset would also be "web-crawled" it would surely not violate the rules shown in this analysis.
- The authors only study end-to-end fine-tuning, but other techniques (e.g. only fine-tuning the last layer, domain adaption, etc.) may yield different results and have not been compared
- The results are not really surprising and generally well-known. but I agree that there is still some value in showing this in actual figures. I am also sure that the community will cite this paper (also due to the exposure of the authors' affiliation), but I am afraid that this work may overshadow all the previous works that studied ImageNet transfer in their own settings. The authors even show even one very thorough investigation [their 69].

**Relation To Prior Work:**

The authors discuss multiple prior works and show a good knowledge of the field. However, many works studied ImageNet transfer in their own settings. Citing them all would be impossible but I am at the same time worried that this work may overshadow individual efforts.

**Summary And Contributions:**

The authors propose to measure how well progress on imageNet correlates with progress on other datasets. In particular, they study the (end-to-end) finetuning of multiple models (CNNs and Transformers) pretrained on ImageNet1k to 6 diverse downstream datasets and measure the correlation between ImageNet accuracy and the performance of the downstream dataset. They concluded that after some initial positive correlations, modern models (after VGG) do not show a notable correlation between the two studied metrics. Additionally, they benchmark augmentation techniques during pretraining and fine-tuning and conclude that augmentations that help on ImageNet do not necessarily improve fine-tuning performance. Finally, the authors study CLIP pre-training and find that it correlates well when controlling for architecture changes.

---

> ### Author Response · Authors · 2023-08-22
>
> Thank you for your time and helpful comments. We are happy that you find this study interesting and relevant, and hope to address some of your concerns.
>
> #### Model Confounders
>
> We agree that the use of new optimizers, augmentations, etc. is a potential confounder and that it is difficult to decouple these from the architecture changes.  However, we believe that this can be somewhat accounted for by comparing downstream performance against ImageNet accuracy, as these changes were made in order to improve ImageNet accuracy. After all, the end goal is to determine whether progress made on ImageNet generalizes to progress towards downstream (real-world) tasks. Nonetheless, in an attempt to partially address this, we did ablations on augmentations in section 4.7 and found that augmentations that improve ImageNet performance do not always improve downstream performance.
>
> #### Web-Scraped vs Real-World Datasets
>
> Our differentiation between web-scraped datasets and the datasets we study is the intention for collecting the data (uploaded for various purposes then web-scraped and assembled, compared to directly collected for a machine learning task), and we believe that this is still compatible with potential explanations of domain gap or distribution shift between ImageNet and the downstream tasks. For example, our discussion regarding the difference in FID scores between the datasets we study and those in Kornblith et al. points to those as potential reasons. But there is probably additional detail beyond that, as CCT20 and Cassava have some resemblance to ImageNet, while HPA and APTOS do not, yet those pairs have different results. While you are right that in the future real-world datasets may be uploaded to the internet and thus become part of the distribution of web-scraped datasets, it would take significant efforts to collect the data, and we believe that such efforts would improve machine learning for real-world datasets.
>
>
> #### Fine-tuning Versus Other Techniques
>
> We choose to study end-to-end fine-tuning because it typically yields the best performance and is most commonly used by practitioners. We agree that investigating other techniques would be interesting follow up work.
>
> #### Clarity and Structure
>
> We thank you for your suggestions regarding clarity and do plan on moving appendix sections (e.g. H, I) into the main paper when there are less space constraints. We would be happy to implement additional suggestions for making it more clear that the transfer learning that we focus on is end-to-end fine-tuning.
>
>
> #### Related Work
>
> Lastly, we acknowledge that this work is only a part of the work done in this area and builds off the work of others. We do not believe that it will overshadow the great work done by others in the past, and hope that our related works section properly acknowledges prior work and shows the relationship between our work and the existing literature.

---

> > ### Comment · Reviewer_ok7Z · 2023-08-22
> > **Response to authors**
> >
> > **Confounders:** The problem definition of this paper is given in L29: *"The goal of our paper is to investigate this possibility **specifically for neural network architecture and their transfer** to real-world data not commonly found on the Internet."*. If the authors don't decouple training from architecture, then they are not solving the problem they defined themselves. The augmentation experiments are a step in the right direction but ultimately only one partial aspect of this.
> >
> > **Web-scraped vs. real-world**: I am sorry, but this differentiation still makes no sense. If the authors want to investigate distribution shifts, then they should do that but the current classification is very far-fetched.
> >
> > After reading the rebuttal, my main concern still stands: This work is incremental and conceptually a collection of findings from hundreds of previous works. There are next to no new insights from this work. Unfortunately, I am now even less convinced that this paper is suitable for publication and have to decrease my score.

---

### Official Review · Reviewer_7tbx · 2023-07-20
**Important study. Would benefit from further analysis.**

**Rating:** 7
**Confidence:** 3
**Correctness:** Yes

**Strengths:**

1.	ImageNet is probably the most popular benchmark in computer vision and is still used as the gold standard for progress on image classification. Papers such as this that question its transferability to real-world applications are crucial.
2.	Experiments are simple and straightforward. I found the paper easy to follow and understand. Decisions are well-motivated.


**Additional Feedback:**

1.	How is the ImageNet column in N.2 so different from correlation results in Figure 1 and Table 2?

Possible points to include in the discussion section:

2.	What is unique about Human protein atlas and CCT that enables transferability?
3.	Could the similarity between categories of CIFAR-10 have something to do with high transferability with ImageNet? Earlier work from some of the same authors (Kornblith et al.) shows that loss function only tunes the last few layers of networks and that earlier layers mainly learn general, non-task-relevant features. Would transferability to real-world datasets from ImageNet improve if a smaller part of the network were used? (i.e. not till the last layer?).


**Clarity:**

Yes, very well written.
I think it will be helpful to mention that ImageNet-pretrained networks are finetuned on the real-world datasets. Currently, all experimental details are included in the appendices, so it is difficult to gather important details such as this from the main paper.

**Documentation:**

Yes, code and datasets are public.

**Ethics:**

No.

**Limitations:**

Yes

**Opportunities For Improvement:**

1.	Performance of networks on 2 of the real-world datasets seems to already be saturated. Area under ROC for the SIIM-ISIC Melanoma dataset is ~95-96%. Accuracy on EuroSAT is ~99%. I wonder if the poor transfer of progress on ImageNet to these datasets could just be because performance on these datasets is already very good, and not because ImageNet itself is a bad benchmark. I’d appreciate authors’ clarification on this.
2.	Possible confound: In Figure 1, evaluated networks mostly lie in the high ImageNet-accuracy regime; hardly 2-3 networks <70% accuracy. Perhaps if more networks in the low accuracy region were added, the authors would see trends similar to what they see across highly accurate networks. I suggest including more poorly-performing networks.
3.    Appendices include interesting analyses that I think would be useful to include in the main paper instead. Particularly the analyses in Appendix N, and Appendix H.


**Relation To Prior Work:**

Yes, very clear.

**Summary And Contributions:**

ImageNet has been and remains a popular dataset for benchmarking neural network image classification. This paper evaluates how well performance improvements on ImageNet transfer to real-world (non-web-scraped) datasets. On testing 19 ImageNet-pretrained networks spanning a wide range of ImageNet accuracies (57-83%) on 6 diverse real-world datasets, the authors find that the transferability of progress was significant when network performance on ImageNet was poor, and small or nil now for models that perform well on ImageNet. Comparing with results from previous work, they hypothesize that ImageNet progress transfers well to web-scraped data but poorly to non-web-scraped data, and support their hypothesis by demonstrating higher FIC-based similarity of ImageNet with web-scraped than non-web-scraped datasets. These results motivate future work to question the utility of ImageNet as the most popular image classification benchmark.

---

> ### Author Response · Authors · 2023-08-22
>
> Thank you for taking the time to review our work. We are happy to see your appreciation for this kind of study, and that you find our work easy to follow.
>
> #### Task Saturation
>
> We agree that it is important to study datasets that are not performance saturated. While EuroSAT may be saturated, the other tasks, including Melanoma, are not because we know of models that can do better on this task. Specifically, the CLIP models we fine-tune in Appendix Figure 8 are better by a non-trivial amount (outside error bars).
>
> #### Low Accuracy Models
>
> We currently have five models below 70% ImageNet accuracy, and have added four more which we present in the table below. An updated figure with these models can be found here: https://drive.google.com/file/d/1NH5xGaBPEhBeMIveOJtQmsLEsnYU9qnn/view?usp=sharing
>
>
> | Model | ImageNet | CCT20 | APTOS | HPA | Melanoma | Cassava | EuroSAT |
> | -------- | -------- | ----------- |  ------- | ------ | -------- | -------- | ------- |
> | tinynet_e | 59.9 | 66.05 | 0.9096 | 0.6189 | 0.9310 | 86.67 | 99.09 |
> | dla46_c | 64.9 | 71.99 | 0.9123 | 0.6265 | 0.9265 | 88.29 | 99.04 |
> | dla46x_c | 66.0 | 70.07 | 0.9088 | 0.6306 | 0.9278 | 88.22 | 98.94 |
> | tinynet_d | 67.0 | 69.99 | 0.9063 | 0.6208 | 0.9365 | 86.35 | 99.07 |
>
> While low accuracy ImageNet models may affect the overall linear fit, it would not significantly affect the high accuracy regime of a polynomial fit. Therefore, we would still find that after some point, progress on ImageNet does not help on some of these tasks.
>
> #### Appendix Structure
>
> Lastly, we agree that some of the appendices would be better in the main paper, but they were actually cut due to space constraints. In the camera-ready version we would like to move at least sections H and I into the main paper.

---

> > ### Comment · Reviewer_7tbx · 2023-08-27
> > **Response to rebuttal**
> >
> > Dear Authors,
> >
> > Thank you for your clear and convincing response to my concerns. My final rating is 7: Accept and I strongly support acceptance of this paper.
> >
> > Thank you,
> > Best
> > Reviewer 7tbx

---

### Official Review · Reviewer_TopE · 2023-07-21
**Paper Review**

**Rating:** 5
**Confidence:** 5
**Correctness:** Maybe not.
**Clarity:** It is clear.

**Strengths:**

- Interesting observation.
- Easy to follow, and well-written.
- Extensive experiments to support their claim.

**Additional Feedback:**

Please resolve my concerns above.

**Documentation:**

Yes.

**Ethics:**

No.

**Limitations:**

They performed several experiments on "real-world datasets". What does "real-world datasets" mean? Even if the explanation can be found in Section 5. "Differences between web-scraped datasets and real-world images", I do not agree with the author's explanation of why the FID between ImageNet and "real-world datasets" is larger than the one between ImageNet and "web-scraped datasets". Why did you use the terminology of "real-world", not "large domain gap"?

Usually, large FID means large domain gaps between two groups as shown in Figure 3. You used medical, satellite, leaf disease, protein, and other datasets. The domain of those datasets is apparently different from the domain of the ImageNet dataset. In such a large domain gap situation, it might be difficult to observe a linear trend shown in Figure 1. What's your opinion on this?

Additionally, [30] used datasets "similar" to the ImageNet dataset, and it implies a small domain gap between the datasets and the ImageNet dataset.

I suggest experimenting with real-world datasets in the same domain as ImageNet such as ObjectNet.

Barbu, Andrei, et al. "Objectnet: A large-scale bias-controlled dataset for pushing the limits of object recognition models." Advances in neural information processing systems 32 (2019).

**Opportunities For Improvement:**

See Limitations.

**Relation To Prior Work:**

Yes.

**Summary And Contributions:**

The paper presents the better model pre-trained using ImageNet is also better in downstream tasks in real-world datasets. The real-world tasks can be classifying images from camera traps or satellites. They provide a figure about transfer performance across models from ImageNet to each of the downstream datasets such as Caltech Camera Traps 20, APTOS, Human Protein Atlas, and others. The authors claim that there seems to be a strong linear trends between ImageNet accuracy and the target metrics, but these trends become less certain when they restrict the models to those above 70% ImageNet accuracy. The paper also includes some experiments on augmentations and CLIP models.

---

> ### Author Response · Authors · 2023-08-22
>
> Thank you for taking the time to review our work and leave thoughtful comments on the potential limitations and framing of our work.
>
> #### On "Real-World Datasets" and Domain Gap
>
> We define "real-world dataset" as a dataset whose images are from the beginning collected for a particular machine learning application. This is in contrast to datasets built via web scraping where the images are first uploaded for a different purpose (e.g., to share photos with friends or to illustrate a blog post), and then later re-purposed for a research dataset. Examples for the latter are  ImageNet and the datasets studied in Kornblith et al., where the images were collected for a variety of reasons unrelated to machine learning, before being web-scraped and assembled for the sake of machine learning. We choose to use this terminology because these are different ways of building datasets and we believe that it is important to ensure that machine learning progress generalizes to performance on both. This choice in terminology is still compatible with your explanation for the FID differences - in fact we may be in agreement, as we note in the discussion section that the FID differences can be attributed to distribution shift.
>
> #### Relationship with ImageNet
>
> ImageNet is used as a benchmark to drive progress in machine learning, which means that when researchers work on models that do better on ImageNet, ideally this progress should transfer to other tasks. We do agree that a good explanation in the differences between Kornblith et al. and our work is the differences in domain gap and distribution shift between ImageNet and the set of datasets each work studies; however, it is still important to study and empirically show these results in order to encourage development of models that do well on these real world tasks for they are still of importance to machine learning. While a larger domain gap makes transfer harder, it is a priori not clear how much of a domain gap transfer learning from ImageNet can handle.
>
> #### Experimenting with ObjectNet
>
> In the case of ObjectNet, while it is a great evaluation dataset and a real world task, it is not studied with fine-tuning and has already been well studied through a robustness perspective (e.g. [1]) - performance on ObjectNet is well correlated with performance on ImageNet. In this work we investigate other real-world datasets that do not necessarily behave like this.
>
> [1] Rohan Taori, Achal Dave, Vaishaal Shankar, Nicholas Carlini, Benjamin Recht, Ludwig Schmidt.``Measuring Robustness to Natural Distribution Shifts in Image Classification.’’ https://arxiv.org/abs/2007.00644

---

### Official Review · Reviewer_EeCa · 2023-07-28
**Straightforward empirical results**

**Rating:** 7
**Confidence:** 4
**Correctness:** I did not identify any issues with co…
**Clarity:** I did not identify any issues with co…

**Strengths:**

**S1) Sufficient and representative models tested**

Testing 19 models ranging from AlexNet through EfficientNet-B4 and ViT-B/16 gives substantial support for the claims made. The authors did extensive hyperparameter tuning and followed standard training procedures.

**S2) Diverse selection of datasets**

The 6 datasets selected represent a reasonably diverse set of tasks and image domains, ranging from camera traps and satellite images to retina images.

**S3) Ablation studies show that augmentations can help where improved ImageNet accuracy does not.**

**Additional Feedback:**

None

**Documentation:**

The paper does not propose any new datasets. The code for reproducing their dataset splits is available on GitHub. However, the code for training their models is not.

**Ethics:**

No concerns.

**Limitations:**

No obvious negative social impacts. Technical limitations are well-addressed in the paper discussion.

**Opportunities For Improvement:**

**O1) Tests need to be done with training from scratch, in addition to initializing with ImageNet weights.**

The experimental results in the paper only consider models whose weights are initialized from weights pre-trained on ImageNet. This leaves 3 major unanswered questions:

- What if these weights are good for web-scraped images, but not real-world images?

- What if the combination of architecture + train-from-scratch procedure is what has led to improved accuracy over the years? After all, models designed for ImageNet are usually trained from scratch.

- How does progress on ImageNet transfer to real-world image datasets (especially remote sensing) that are not just RGB images?


Regarding the 3rd point above, the land-cover classification task in SustainBench ([https://sustainlab-group.github.io/sustainbench/docs/datasets/sdg15/land_cover_representation.html](https://sustainlab-group.github.io/sustainbench/docs/datasets/sdg15/land_cover_representation.html)) uses 4-band images (RGB + near infrared). For tasks like this, using ImageNet weight initialization doesn’t make sense.

Thus, I would like to see the following experiments:

- Models with weights trained from random initialization, instead of pre-trained on ImageNet.

- (time-permitting) Testing on a non-RGB dataset.


**O2) The paper does not sufficiently address the issue of dataset size.**

The authors write that “Training set sizes are similar between our study and that of Kornblith et al. [30] and thus also do not seem to play a major role.” However, both Kornblith et al. (2019) and this manuscript only trained on datasets with ≤50K training examples. In contrast, ImageNet itself is much larger (~1M examples). The paper does not address the possibility that architectures designed for ImageNet will work well on large-scale real-world (not web-scraped) datasets.

The the land-cover classification task in SustainBench that I mentioned above could serve as one of these large-scale real-world datasets. It has 200K examples.

**O3) Does progress on ImageNet transfer to improvements in calibration?**

The manuscript currently only tests models’ accuracy, QWK, or macro F1 score, but it doesn’t show whether progress on ImageNet transfers to other metrics such as calibration. For example, Galil et al. (2023 - [https://openreview.net/forum?id=p66AzKi6Xim](https://openreview.net/forum?id=p66AzKi6Xim)) found that ViT models not only have better accuracy but also better uncertainty estimates on ImageNet. How does this apply to transfer learning tasks?

**Relation To Prior Work:**

The paper very clearly highlights the differences between itself and Kornblith et al. (2019), which tested downstream accuracy on other web-scraped computer vision tasks.

**Summary And Contributions:**

The authors test whether 19 models trained on ImageNet transfer well to 6 different “real-world” image classification datasets. The authors claim that modern “ImageNet-motivated architecture improvements after VGG resulted in little to no progress” on these real-world datasets and provide evidence suggesting that this phenomenon may be due to the measurable distribution shift between web-scraped datasets (like ImageNet) and real-world datasets.

My understanding of the main contributions are:

C1) Transfer learning tests of 19 models pre-trained on ImageNet for 6 real-world image classification tasks show that progress on ImageNet beyond a certain point only yields marginal performance gains (if any) on real-world image classification tasks.

C2) The authors provide evidence suggesting that one potential reason for the lack of transfer-ability is the distribution shift between web-scraped datasets (like ImageNet) and real-world images.

C3) The paper proposes using average performance on the 6 real-world image classification datasets as a more realistic benchmark for future representation learning research.

---

> ### Author Response · Authors · 2023-08-22
>
> Thank you for your time and suggestions for improvement. We are happy to see that you find the range of models tested representative, and the datasets that we study diverse. To help with future research, we have added sample training code to the same github link that contains the data references and splits.
>
> > O1) Tests need to be done with training from scratch, in addition to initializing with ImageNet weights.
>
> While in a different setting, He et al. [1] find that the main practical benefit of pre-training of using pre-trained weights is for faster convergence time, and that transfer performance typically is better than or at least the same as the performance of training from scratch; initial experiments on CCT20 and APTOS suggested the same, so in this work we focused on the transfer learning aspect. Furthermore, practitioners typically start from a pre-trained model and use transfer learning because training from scratch is more expensive. However, we agree that studying this in depth would be interesting follow up work because of the potentially different nature of these datasets. Because of this decision to focus on transfer learning, non-RGB datasets are unfortunately out of scope for this study.
>
> > O2) The paper does not sufficiently address the issue of dataset size.
>
> Regarding the concern of dataset size, collecting real world datasets is a lot of work, so it typically is not practical to scale up such datasets to ImageNet size. Nonetheless, if the goal of using ImageNet as a benchmark is to improve vision models overall, just like how we would like to see that better ImageNet models are also better models for real world tasks, better ImageNet models should still perform better when fine-tuned on smaller datasets.
>
> > O3) Does progress on ImageNet transfer to improvements in calibration?
>
> Regarding calibration, the paper linked seems to suggest that pre-trained models do not perform significantly better than models trained from scratch. Since these models are not designed with calibration in mind, we would assume that there would not be too big of a difference when fine-tuning on real-world datasets when compared to web-scraped datasets, but this would definitely be interesting follow up work.
>
> [1] Kaiming He, Ross Girshick, Piotr Dollar. ``Rethinking ImageNet Pre-training’’ https://arxiv.org/pdf/1811.08883.pdf

---

> > ### Comment · Reviewer_EeCa · 2023-08-29
> > **Paper needs to clarify scope of contributions**
> >
> > I thank the authors for the thoughtful responses. Given that the authors did not run new experiments for training from scratch or on larger datasets, the final version of the paper should make very clear that:
> >
> > 1) The claims in the paper are only valid for RGB images, and not other images such as multi-spectral satellite images.
> >
> > 2) The claims in the paper are only valid for small datasets (≤50K training examples).
> >
> > As long as the authors explicitly acknowledge these two statements in their final paper, I would be happy to keep my rating of 7.

---

### Official Review · Reviewer_c3Q1 · 2023-08-02
**Interesting Problem but Benchmarks can be Improved**

**Rating:** 5
**Confidence:** 5

**Strengths:**

* This paper tries to benchmark an important but not well-defined research problem in computer vision, i.e., the transferring abilities to real-world applications of ImageNet-trained models. And this paper provides comprehensive results across various image classification scenarios and model architectures.

* The experiment results are well-arranged, and the authors provide some interesting findings which were not well-studied.


**Additional Feedback:**

No more questions.

**Clarity:**

* Overall, the manuscript is easy to follow and well-organized.

* There are some minor points can be improved. For example, the author can include more properties of the selected six datasets in Table 1 (or using a new one) to leave out more space for some valuable findings in the main text. Meanwhile, the authors should use high-quality images in the main text, e.g., using PDF format for Figure 3 instead of the low-quality version.

**Correctness:**

Some conclusions might not be sufficiently verified or demonstrated. For example, the authors concluded that the parameter size of the model is not correlated to transferring performances on real-world datasets. However, this conclusion is conducted by limited scaling-up experiments (e.g., classical architectures like VGG and ResNet). The authors should evaluate modern architectures with wide parameter sizes (e.g., ViT and ConvNeXt) to verify the correctness. I suggest the authors modify the conclusions or conduct verification experiments to make them more reliable and comprehensive.

**Documentation:**

The authors have not provided GitHub links or online documents that contain users’ guidance and benchmark results. I suggest the authors attach relevant material. Meanwhile, the benchmark results in the appendix are comprehensive and provide downloaded links.

**Limitations:**

* From the perspective of the selected datasets, I have three concerns. Firstly, the authors should study the overlap between ImageNet and the selected six datasets, e.g., analyzing the class and image statistics. This information is closely related to whether this benchmark can truly reflect the transferring abilities of ImageNet models. Secondly, I wonder why the authors were not taking some well-benchmarked classical image classification datasets into consideration, e.g., CUB-200-2011 [1] and Place205 [2]. These existing datasets (not performance-saturated) are widely used by supervised and self-supervised learning methods [3] to evaluate the transferring abilities to downstream classification tasks. However, the selected datasets might not be popular and well-benchmarked by existing techniques of image classifications (e.g., network architectures, data augmentations [4, 5], and regularization strategies [6]). Thirdly, the class numbers of selected datasets are small to some extent, which might not be as difficult as some existing benchmark datasets and ImageNet.

* From the perspective of evaluated models, I also have two concerns. Firstly, the authors mainly evaluate classical models available at the torchvision website (ResNet variants and light-weight CNNs) while not taking popular modern architecture proposed from 2021 to 2023 into consideration (e.g., Swin Transformer [7], MLP-Mixer variants [8], and Metaformer baselines [9]). Please refer to Timm [10] for more pretrained models on ImageNet. Secondly, the authors evaluate many lightweight models while overlooking recently proposed Transformer models and self-supervised models (e.g., contrastive learning [3] and masked image modeling [11]). I wonder whether task-agnostic pre-training can also bring essential improvements as the task-specific data augmentations to transferring of ImageNet trained models.

* The open-source code or project of this paper should be provided, which is a sufficient condition for a general benchmark. I have not found the code or project in supplementary or online. Moreover, the models and logs of experiment results should be made available to the community in order to reproduction and further research.

### Reference
[1] Wah et al. The Caltech-UCSD Birds-200-2011 Dataset. 2011.

[2] Zhou et al. Learning Deep Features for Scene Recognition using Places Database. NeurIPS, 2014.

[3] Ting Chen, et al. A Simple Framework for Contrastive Learning of Visual Representations. ICML, 2020.

[4] Hongyi Zhang, et al. mixup: Beyond Empirical Risk Minimization. ICLR, 2018.

[5] Zicheng Liu et al. AutoMix: Unveiling the Power of Mixup for Stronger Classifiers. ECCV, 2022.

[6] Ross Wightman, et al. ResNet strikes back: An improved training procedure in timm. 2021.

[7] Ze Liu, et al. Swin Transformer: Hierarchical Vision Transformer using Shifted Windows. ICCV, 2021.

[8] Ilya Tolstikhin, et al. MLP-Mixer: An all-MLP Architecture for Vision. NeurIPS, 2022.

[9] Weihao Yu, et al. MetaFormer Baselines for Vision. 2022.

[10] Ross Wightman. PyTorch Image Models, https://github.com/rwightman/pytorch-image-models. 2019.

[11] Kaiming He, et al. Masked Autoencoders Are Scalable Vision Learners. CVPR, 2022.

**Opportunities For Improvement:**

Please refer to issues in Limitations, Correctness, and Clarity. I think this manuscript and benchmark can be further improved before being accepted.

**Relation To Prior Work:**

* The authors should include more existing image classification datasets of downstream tasks (e.g., fine-grained classification like CUB-200-2011 and iNatualist variants). Since some existing datasets also satisfy the three criteria the authors proposed, these datasets have already been studied by the community.

* More papers on recently proposed network architectures and self-supervised learning approaches should be discussed. Since recently-proposed architectures usually consider the transferring abilities of the (pre)-trained models.

**Summary And Contributions:**

This paper proposes a new benchmark that investigates the transferring abilities of models trained on ImageNet to real-world datasets on six selected image classification datasets. Concretely, the selected datasets satisfy three criteria (i.e., diverse data sources, application relevance, and availability of various baselines), and most of them were collected with the goal of solving real-world tasks. Based on the evaluation results, the authors summarize many interesting conclusions, e.g., data augmentations improve performances for certain tasks while architectures do not. The proposed benchmarks are likely to encourage the community to design more comprehensive methods for real-world classification tasks.

---

> ### Author Response · Authors · 2023-08-22
>
> Thank you for taking the time to leave us thoughtful feedback. We are happy to see your interest in the scope of the problem we choose to study.
>
> #### Dataset Selection
>
> We provide statistics of our datasets in Table 1, mention potential overlap with ImageNet classes in the main experiments section, and compare them with the datasets studied by Kornblith et al. in the discussion section. If there are additional analyses that you have in mind, we would be happy to conduct them.
>
> Regarding the selection of the datasets we choose to study, we choose predominantly datasets from Kaggle because they are real world datasets where the images are collected for the purpose of doing machine learning. While the datasets you mention (CUB-200-2011, Place205) are definitely interesting, the images were initially uploaded to the web for a variety of purposes, before being web scraped (e.g. from Flickr) and then collected for machine learning. Despite some of our chosen datasets being less studied in the academic setting, machine learning practitioners from the Kaggle community have spent lots of effort on improving methods used on these tasks, which provides benchmarks for comparison.
>
> Task difficulty is definitely a valid concern, but while the dataset sizes are relatively small compared to ImageNet, Kornblith et al. found that better ImageNet models do indeed transfer better on other datasets of similar sizes. From a pre-training perspective, Huh et al. [1] find that pre-training dataset size and class size do not significantly affect transfer performance. Furthermore, we believe the majority of these datasets are not performance saturated yet - we found that simply fine-tuning a CLIP model pre-trained on a much larger dataset does better than fine-tuning on ImageNet pre-trained models.
>
> #### Modern Models
>
> We appreciate you bringing to our attention the importance of benchmarking on the latest set of models. In addition to our existing ConvNext experiments, we have run additional experiments on a SWIN base model, and a CAFormer, with the results below for AdamW + cosine schedule. We will run the full hyperparameter grid and update tables and plots for the next revision. Note that MLP Mixer does not have IN1k pre-trained weights on timm (only IN21k + IN1k).
>
> | Model | ImageNet | CCT20 | APTOS | HPA | Melanoma | Cassava | EuroSAT |
> | -------- | -------- | ----------- |  ------- | ------ | -------- | -------- | ------- |
> | SWIN Base | 83.6 | 76.79 | 0.9317 | 0.7369 | 0.9647 | 88.43 | 99.44 |
> | CAFormer B36 | 85.5 | 79.36 | 0.9261 | 0.6939 | 0.9669 | 88.92 | 99.41 |
>
> Here are the updated correlations and slopes for models over 70% (Table 2):
>
> | Dataset      | Correlation | Slope |
> | -------- | -------- | ----------- |
> | CCT 20      | 0.43 | 0.28 |
> | APTOS      | 0.23 | 0.03 |
> | HPA           | 0.39 | 0.39 |
> | Melanoma | 0.63 | 0.08 |
> | Cassava    | 0.31 | 0.06 |
> | EuroSAT    | 0.36 | 0.02 |
>
> While the slopes have gone up slightly, progress on downstream tasks still significantly lags behind progress on ImageNet for some of the datasets. An updated figure with these models can be found here: https://drive.google.com/file/d/1NH5xGaBPEhBeMIveOJtQmsLEsnYU9qnn/view?usp=sharing
>
>
> Though the majority of our models are convolutional based models, we do include transformer variants (DeiT, ViT). While our overall focus is on studying the traditional supervised classification training and procedure, we do investigate CLIP models (contrastive learning), though the focus there is on the differences in pre-training data. Models trained on these other methods (contrastive, self-supervised) often rely on pre-training on larger datasets before fine-tuning on the downstream task, so these are not directly comparable to the supervised models pre-trained on ImageNet.
>
> #### Open-Source and Reproducibility
>
> We agree that open-source code and reproducibility are important. Our paper contains a link to https://github.com/mlfoundations/imagenet-applications-transfer, which contains references for downloading the data, as well as torch code for loading the splits for each of the datasets we study. In the revision we will make this clearer by adding it to the abstract or the end of the introduction. The appendix also contains training hyperparameters, so we believe that all experiments in the paper are reproducible. Additionally we have added sample training code to the above link, which we hope eases future replication and follow up studies.
>
> [1] Minyoung Huh, Pulkit Agrawal, Alexei A. Efros. ``What makes ImageNet good for transfer learning?’’ https://arxiv.org/abs/1608.08614

---

### Comment · Area_Chair_4Dkb · 2023-08-27
**Please respond to author rebuttals**

Dear reviewers c3Q1, EeCa, TopE and 7tbx,

Would you please check in, to confirm whether you have red the rebuttal of the authors, and whether this has changed the way you rate the paper?

Best,
The AC

---

### Decision · Program_Chairs · 2023-09-22

**Decision:**

Accept (Poster)

**Comment:**

The authors aim to provide an answer to the question, does progress on ImageNet transfer to real-world datasets?

While all reviewers have submitted their reviews on time, not all reviewers actively participated in the discussions, making this a complicated paper to assess. Reviewers commit to not only review the paper, but to also participate in the author discussion, which was neglected by some, despite additional encouragements. While based on the current scores the paper would be below the acceptance threshold, it might be good to assess the individual concerns.

1) Concerns regarding the availability of the open source code - based on my personal assessment this seems addressed by the authors, and no further discussion was made by the reviewer
2) Concerns regarding the term of real-world datasets - This point does not seem addressed to the satisfaction of the reviewer
3) Concerns regarding the training process and exact architecture - This point does not seem addressed to the satisfaction of the reviewer
4) Concerns regarding the transferability between the datasets and ImageNet - based on my personal assessment this seems addressed by the authors, and no further discussion was made by the reviewer
5) Concerns regarding the lack of already often-used datasets such as CUB200 or ObjectNet.
6) Concerns to the size of the datasets.
7) Concerns regarding the used models, and more modern models could have been chosen.

When it comes to point 5 6 and 7, while being valid concerns, it is my assessment that by these kind of remarks are always applicable to any type of study. While I do believe that the authors could have taken a more pro-active attitude and include these models nonetheless, I believe that these are not the points that should sway the discussion on themselves towards a negative outcome.

The points regarding the term real-world dataset is a valid one, that should be handled serious. However, I would like to point the reviewer towards the discussion on "Differences between web-scraped datasets and real-world images". I support the statement of both reviewers that brought up this point (of which 1 participated in the discussion), but acknowledge that several terms and definitions can have different meanings in different areas of science. In my opinion, adding this discussion aids to a better understanding of what the authors mean, and in my opinion, this qualifies as a reasonable response to the valid concern.

All together, my assessment seems that this is valuable work with many implications for science, but that it comes with certain limitations. The authors could and should have done a better job addressing these. All together, my recommendation would be to accept the paper nonetheless for a poster presentation, so that the rest of the community can learn from and build upon these results.